# VALUE BONUSES USING ENSEMBLE ERRORS FOR EXPLORATION IN REINFORCEMENT LEARNING

## ABSTRACT

Optimistic value estimates provide one mechanism for directed exploration in reinforcement learning (RL). The agent acts greedily with respect to an estimate of the value plus what can be seen as a *value bonus*. The value bonus can be learned by estimating a value function on *reward bonuses*, propagating local uncertainties around rewards. This approach, however, only increases the value bonus for an action retroactively, after seeing a higher reward bonus from that state and action. Such an approach does not encourage the agent to visit a state and action for the first time. In this work, we introduce an algorithm for exploration called Value Bonuses with Ensemble errors (VBE), that maintains an ensemble of random action-value functions (RQFs). VBE uses the errors in the estimation of these RQFs for designing value bonuses that provide first-visit optimism and deep exploration. The key idea is to design the rewards for these RQFs in such a way that the value bonus can decrease to zero. We show that VBE outperforms Bootstrap DQN and two reward bonus approaches (RND and ACB) on several classic environments used to test exploration and provide demonstrative experiments that it learns faster in several Atari environments.

## 1 INTRODUCTION

A typical approach to incorporate exploration into a value-based reinforcement learning (RL) agent is to obtain optimistic value estimates. The agent takes greedy actions according to this optimistic value estimate, leading it to take actions that look good either because they have high uncertainty or because the action is actually high value. This approach has been well-developed for the contextual bandit setting, with a variety of algorithms and theoretical results on optimality (Li et al., 2010; Abbasi-Yadkori et al., 2011). Understanding is growing about how to soundly extend these ideas to reinforcement learning, though the theoretical results on estimating and using optimistic values are limited to the linear function approximation setting (Grande et al., 2014; Osband et al., 2016; Abbasi-Yadkori et al., 2019; Wang et al., 2019).

Though the theory is difficult to extend, there has been a concerted effort to develop and empirically evaluate such optimistic value estimation approaches for the deep RL setting. Bootstrap DQN with priors, for example, maintains an ensemble of action-values, which reflect uncertainty in the value estimates (Osband et al., 2018; 2019). It takes a Thompson sampling approach—which can be seen as optimistic—by sampling one action-value in the ensemble and following it for an entire episode. Another common approach to obtain optimistic value estimates employs the usage of *reward bonuses* (Bellemare et al., 2016; Ostrovski et al., 2017; Burda et al., 2019; Ash et al., 2022). A reward bonus, reflecting uncertainty with respect to the transition, is added to the reward, increasing the estimated value proportionally for the corresponding states and action.

Most works, however, eschew these directed exploration approaches in favor of simpler, undirected exploration approaches like $\epsilon$-greedy. One potential reason for this is that reward bonus approaches do not encourage *first-visit optimism*. They encourage revisiting a state, if the reward bonus was high in that state; namely, they retroactively reason about uncertainty of states they have seen. The reward bonus cannot encourage visiting a state for the first time. Bootstrap DQN with priors (BDQN), on the other hand, does not have this issue, using fixed additive priors to provide first-visit optimism. Unlike reward bonuses, though, BDQN is more onerous to use. It requires completely changing the algorithm to one that maintains and updates an ensemble, and making key choices like how often to

follow one of the value functions in the ensemble before switching. Recent work suggests it is key to have a large ensemble for BDQN (Janz et al., 2019; Osband et al., 2023). Epinets (Osband et al., 2023) can match the performance of BDQN with much less compute, but are arguably even more onerous to implement than BDQN. Our goal is to develop an easy-to-use exploration approach for deep RL, that can easily be added to an existing algorithm, making it less onerous to displace the default $\epsilon$-greedy approach.

To do so, we explore how to directly estimate a *value bonus*. The agent acts greedily according to the value estimate plus this separate value bonus $b$, namely $\operatorname{argmax}_a q(s,a) + b(s,a)$. The value bonus should ideally represent the uncertainty for that state and action. Though this may be the first time this term is used,[1] there are some works that estimate value bonuses. One simple approach is to separate out the reward bonuses and learn them with a second value function, as was proposed for RND (Burda et al., 2019) and later adopted by ACB (Ash et al., 2022). This approach, however, still suffers from the fact that reward bonuses are only retroactive, and the resulting $b$ is unlikely to be high for unvisited states and actions. For the contextual bandit setting, the ACB algorithm actually directly estimates the value bonus using the maximum over an ensemble of functions, which is high for unvisited states and actions; but the extension to deep RL with reward bonuses loses this first-visit optimism. UCLS (Kumaraswamy et al., 2018) and UBE (O'Donoghue et al., 2018; Janz et al., 2019) both directly estimate value bonuses, but are limited to linear function approximation. Dora (Choshen et al., 2018) uses value bonuses that are inversely proportional to visitation counts, which is again difficult to extend to the general function approximation setting.

In this work, we introduce a new approach to obtain value bonuses for reinforcement learning, with an algorithm we call Value Bonuses with Ensemble errors (VBE). Similarly to ACB, we use a maximum over an ensemble, but directly use that maximum as the value bonus, rather than indirectly through reward bonuses. The idea is to sample a random action-value function (RQF)—such as a random neural network—and extract the implicit random reward function underlying this RQF target. The RQF predictor in the ensemble is updated using temporal difference learning on this random reward. Because the RQF target is sampled from the same function class as the RQF predictor, the error can eventually reduce to zero, allowing the value bonus shrink to zero. These value bonuses are learned separately from the main action-values, and so can be layered on top of many algorithms. In our experiments, for example, we simply use Double DQN (Van Hasselt et al., 2016), and modify the step where the agent selects an action from $\epsilon$-greedy to instead taking the greedy action in the value estimate plus the value bonus. We show that this simple approach is an effective, and scalable method for exploration that improves sample efficiency of learning in a range of domains: from hard exploration gridworlds, to image-based Atari domains.

## 2 BACKGROUND

We focus on the problem of an agent learning optimal behaviour in an environment, whose interaction process is modelled as a Markov Decision Process (MDP). A MDP consists of $(\mathcal{S}, \mathcal{A}, P, r, \gamma)$ where $\mathcal{S}$ is the set of states; $\mathcal{A}$ is the set of actions; $P : \mathcal{S} \times \mathcal{A} \times \mathcal{S} \to [0, \infty)$ provides the transition probabilities; $r : \mathcal{S} \times \mathcal{A} \times \mathcal{S} \to \mathbb{R}$ is the reward function; and $\gamma : \mathcal{S} \times \mathcal{A} \times \mathcal{S} \to [0, 1]$ is the transition-based discount function which enables either continuing or episodic problems to be specified (White, 2017). On each step, the agent selects action $A_t$ in state $S_t$, and transitions to $S_{t+1}$, according to $P$, receiving reward $R_{t+1} \stackrel{\text{def}}{=} r(S_t, A_t, S_{t+1})$ and discount $\gamma_{t+1} \stackrel{\text{def}}{=} \gamma(S_t, A_t, S_{t+1})$.

For a policy $\pi : \mathcal{S} \times \mathcal{A} \to [0, \infty]$, the value for taking action $a$ in state $s$ is the expected discounted sum of future rewards, with actions selected according to $\pi$ in the future,

$$q^\pi(s,a) = \mathbb{E}_\pi \left[ R_{t+1} + \gamma_{t+1} q^\pi(S_{t+1}, A_{t+1}) \Big| S_t = s, A_t = a \right]$$

where $\mathbb{E}_\pi$ means that actions are selected according to $\pi$ in the expectation. The policy $\pi$ can be progressively improved by making it greedy in $q^\pi(s,a)$, then updating the action-values for the new policy, then repeating until convergence.

---

[1]Usually, $b$ would be called a confidence interval, with $q(s,a) + b(s,a)$ an upper confidence bound. However, we do not use that term here, because for the heuristics we use, it is not clear we get a valid upper confidence bound. Instead, it is a bonus added to the value when deciding which action looks promising.

In practice, these steps are approximated. The action-values $q^\pi$ are approximated using $q_w$ parameterized by $w \in \mathcal{W} \subset \mathbb{R}^d$. One algorithm to estimate $q_w$ is Double DQN (DDQN). DDQN is an off-policy algorithm, meaning that it uses a different behavior policy $\pi_b$ to select actions from the policy it evaluates, which is greedy in $q_w$. This algorithm uses a target network $q_{\tilde{w}}$ for bootstrapping, giving the following update for one transition $(s, a, r, s', \gamma)$:

$$w \leftarrow w + \eta \delta \nabla q_w(s, a) \quad \text{for } \delta \stackrel{\text{def}}{=} r + \gamma q_{\tilde{w}}(s', \operatorname*{argmax}_{a'} q_w(s', a')) - q_w(s, a) \tag{1}$$

The behavior policy is typically defined to be $\epsilon$-greedy in $q_w$, but can be any policy that promotes exploration. In this work, we consider an alternative choice for the behavior policy: one that uses a value bonus $b$, $\pi_b(s) = \operatorname{argmax}_a q_w(s, a) + b(s, a)$. The value bonus should reflect uncertainty in the action-value estimate, encouraging the behavior policy to take an action in a state if it has high uncertainty. It might have high uncertainty if $(s, a)$ is quite different from what it has seen before—meaning it has never been visited—or because the agent has not yet visited it sufficiently often to be certain about its value. The focus of this work is a new approach for obtaining $b$ for the deep RL setting.

## 3 Value Bonuses with Ensemble Errors

In this section, we first motivate why we use an ensemble of value functions, rather than simply using supervised learning for the ensemble. We then discuss how to appropriately define the rewards for the ensemble value functions, and finally provide the VBE algorithm that uses this ensemble.

The most straightforward approach to get an error from an ensemble is to use a random target, as is done in RND. For an ensemble of size $k$, we can generate random neural networks $f_1, \ldots, f_k$ and update the learned functions $\hat{f}_1, \ldots, \hat{f}_k$ in the ensemble using a squared error: for each $(s, a)$, update each $\hat{f}_i$ using loss $(f_i(s, a) - \hat{f}_i(s, a))^2$. The value bonus for any $(s, a)$ can be set to

$$b(s, a) \doteq \max_{i \in [k]} |\hat{f}_i(s, a) - f_i(s, a)| \tag{2}$$

Ciosek et al. (2020) show that fitting random prior functions serve as a computationally tractable approach towards estimating uncertainty in the supervised learning setting. Unfortunately, in the reinforcement learning setting, this is likely to concentrate too quickly, and will not do what has been called deep exploration (Osband et al., 2019). We want the agent to reason not just about uncertainty for this state and action, but also about the uncertainty of the state that it leads into.[2]

Instead, we want an ensemble of value functions that are more likely to promote deep exploration. More specifically, we want to generate random rewards $r_i$ for each $f_{w_i}$, where the $f_{w_i}$ are updated using standard temporal difference learning bootstrapping approaches. We want the learning dynamics for these value functions to resemble the primary value function, so that they learn at a similar timescale and are more likely converge to zero once the primary value function has also converged.

We need to define rewards and target functions that are consistent with each other and that allow us to easily measure the errors. Consider if we again do the simplest thing: generate a random neural network $r_i$ for each $f_{w_i}$. Let us assume for now that we have a fixed policy, $\pi$. First, it is not clear how we would actually measure the error since we do not know the true value function $f_i$, namely the expected return using $r_i$ under policy $\pi$. Further, this true value function may not be representable by $f_{w_i}$.

Instead, our proposed approach is to generate a random action-value function (RQF) $f_i$, and then define rewards consistent with that $f_i$. Define the stochastic ensemble reward from $(S_t, A_t)$ to be

$$R_{i,t+1} \stackrel{\text{def}}{=} f_i(S_t, A_t) - \gamma_{t+1} f_i(S_{t+1}, A_{t+1}), \tag{3}$$

where $A_{t+1} \sim \pi(\cdot | S_{t+1})$ and $\gamma_{t+1} \stackrel{\text{def}}{=} \gamma(S_t, A_t, S_{t+1})$ is defined by the environment. Further, by definition, the action-values of the random prediction function is:

$$q_i^\pi(s, a) \stackrel{\text{def}}{=} \mathbb{E}_\pi \left[ R_{i,t+1} + \gamma_{t+1} q_i^\pi(S_{t+1}, A_{t+1}) \big| S_t = s, A_t = a \right]. \tag{4}$$

We show in the following proposition that $q_i^\pi = f_i$.

---

[2]Note that RND do not use these errors directly for exploration. Instead, they used them as reward bonuses, which can retroactively promote deep exploration, with the issue that they do not promote first-visit optimism.

**Proposition 1** *For all $i \in [k]$, we have $q_i^\pi = f_i$.*
**Proof:**

$$
\begin{aligned}
q_i^\pi(s,a) &= \mathbb{E}_\pi \left[ R_{i,t+1} + \gamma_{t+1} q_i^\pi(S_{t+1}, A_{t+1}) \big| S_t = s, A_t = a \right] \\
&= \mathbb{E}_\pi \left[ R_{i,t+1} + \gamma_{t+1} R_{i,t+2} + \gamma_{t+1}\gamma_{t+2} q_i^\pi(S_{t+2}, A_{t+2}) \big| S_t = s, A_t = a \right] \\
&= \mathbb{E}_\pi \Big[ [f_i(s,a) - \gamma_{t+1} f_i(S_{t+1}, A_{t+1})] + \gamma_{t+1}[f_i(S_{t+1}, A_{t+1}) - \gamma_{t+2} f_i(S_{t+2}, A_{t+2})] \\
&\qquad + \gamma_{t+1}\gamma_{t+2} q_i^\pi(S_{t+2}, A_{t+2}) \big| S_t = s, A_t = a \Big] \\
&= \mathbb{E}_\pi \Big[ [f_i(s,a) \underbrace{-\gamma_{t+1} f_i(S_{t+1}, A_{t+1})]}_{\text{cancels}} \underbrace{+\gamma_{t+1} f_i(S_{t+1}, A_{t+1})}_{\text{cancels}} \\
&\qquad - \gamma_{t+1}\gamma_{t+2} f_i(S_{t+2}, A_{t+2})] + \gamma_{t+1}\gamma_{t+2} q_i^\pi(S_{t+2}, A_{t+2}) \big| S_t = s, A_t = a \Big]
\end{aligned}
$$

We can keep unrolling this, and these terms will continue to telescope, leaving only the first term $f_i(s,a)$, completing the proof. ∎

Therefore, updating $f_{w_i}$ with rewards $r_i$ should converge to $q_i^\pi$—and so to $f_i$—because $f_i$ is in the function class of $f_{w_i}$. This convergence ensures the value bonuses go to zero, which is desired if we want the agent to stop exploring and converge to the greedy policy. Even with a fixed policy, however, this convergence will only occur under certain conditions. Primarily, the failure would be that $f_{w_i}$ gets stuck in a local minima or even that it diverges, due to know issues with temporal difference (TD) learning algorithms combined with neural networks and with off-policy update.

There is fortunately a large (and growing) literature understanding the convergence behavior of TD algorithms. Under linear function approximation, we know least-squares TD converges at a rate of $1/\sqrt{T}$ to the global solution, even under off-policy sampling (Tagorti & Scherrer, 2015). With the advent of theory for overparameterized networks, TD with a particular neural network function class has been shown to converge to the global solution, under on-policy sampling (Cai et al., 2019). In general, we know that a class of modified TD algorithms, called gradient TD methods, converge even under off-policy sampling and nonlinear function approximation (Dai et al., 2017; Patterson et al., 2022). Convergence under off-policy sampling is key in our setting, because the behavior policy is optimistic but the target policy may be greedy. We expect that under certain conditions on the neural network it might be possible to say that these gradient TD methods converge to global solutions, though to the best of our knowledge, no such work yet exists. We provide a more complete discussion in Appendix A of how this existing theory on convergence of TD applies to our setting.

## 4 USING THE ENSEMBLE OF VALUE FUNCTIONS

We provide the Value Bonuses with Ensemble Errors (VBE) algorithm in this section. We provide pseudocode, in Algorithm 1, for the case where the base algorithms is Double DQN, but it is possible to swap in many different off-policy value-based algorithms. Even actor-critic, which explicitly maintains a critic $q_w$, could easily incorporate the value bonuses by using instead an optimistic critic. For the purposes of this paper, however, we restrict our focus to Double DQN.

The ensemble value functions are updated on the same target policy as Double DQN, namely the greedy policy in $q_w$. This choice comes from the fact that we want to understand uncertainty in the values for the target policy. The update is similar to Double DQN, except the actions are sampled according to $q_w$ rather than $f_{w_i}$, and we use the ensemble reward $r_i$ defined above in Equation (3):

$$
w_i \leftarrow w_i + \eta \delta_i \nabla f_{w_i}(s,a) \quad \text{for } \delta \overset{\text{def}}{=} r_i + \gamma f_{\tilde{w}_i}(s', \underset{a'}{\arg\max}\, q_w(s', a')) - f_{w_i}(s,a) \tag{5}
$$

On each step, we only update one RQF predictor. Updating the entire ensemble is expensive, and arguably unnecessary. There are multiple ways to control the magnitude of the value bonus, and how quickly it decays. One way is the size of the ensemble, where the larger the ensemble, the more slowly this bonus should decay. Updating each RQF predictor less frequently, however, will also cause the bonus to decay more slowly. It both allows us to make the ensemble smaller, and ensure that regardless of the ensemble size, the computation per-step is simply double that of Double DQN: one update to the main value function and one update to an RQF predictor.

---

**Algorithm 1** Value Bonuses with Ensemble Errors (VBE)

---

1: **Parameters:** ensemble size $k$, bonus scale $c$, target net update frequency $\tau$, batch size $m$
2: Initialize empty buffer: $B \leftarrow \emptyset$, action-value function: $q_w$, target RQFs: $f_i, \ldots f_k$, predictor RQFs: $f_{w_1}, \ldots, f_{w_k}$, and target networks: $q_{\tilde{w}}, f_{\tilde{w}_1}, \ldots, f_{\tilde{w}_k}$
3: **Optimistic behavior policy:** $\pi_b(s) \leftarrow \mathrm{argmax}_{a \in \mathcal{A}} \, q_w(s, a) + c \, b(s, a)$
   where $b(s, a) \leftarrow \max_{i \in [k]} |f_{w_i}(s, a) - f_i(s, a)|$
4: Get the initial state $s_0$
5: **for** environment interactions $t = 0, 1, \ldots$ **do**
6:     Take action $a \leftarrow \pi(s_t)$ and observe $r_{t+1}, s_{t+1}, \gamma_{t+1}$
7:     Add $(s_t, a_t, r_{t+1}, s_{t+1}, \gamma_{t+1})$ to the buffer $B$
8:     // Update the action-values using the Double DQN update
9:     Sample a mini-batch and use update in Equation (1) to update $q_w$
10:     // Update one randomly select ensemble value function
11:     Sample $i$ from $[k]$ uniform randomly
12:     Sample a mini-batch from $B$ and update $f_{w_i}$ using Equation (5) where
       for each $(s, a, r, s', \gamma)$ we replace $r$ with $r_i \stackrel{\text{def}}{=} f_i(s, a) - \gamma f_i(s', \mathrm{argmax}_{a' \in \mathcal{A}} q_w(s', a'))$
13:     **if** $t + 1 \mod \tau == 0$ **then**
14:         $\tilde{q}_w \leftarrow q_w$ and for all $i$, $f_{\tilde{w}_i} \leftarrow f_{w_i}$
15:     **end if**
16: **end for**

---

We can again ask what happens to our value bonuses in VBE. Ideally, they eventually converge to zero, with the action-values converging and the behavior and target policies both converging to a greedy policy. This scenario goes beyond the convergence conditions discussed above in Section 3 for fixed policies. In VBE, both our behavior policy and target policy are changing with time. Unfortunately, theory around TD does not address this scenario. There are some results for a fixed behavior policy for double Q-learning under linear function approximation (Zhao et al., 2021), or for a variant of DQN with a fixed dataset (Wang & Ueda, 2022). The issue with a changing behavior policy is that it changes the relative importance of states in the objective, and so the best value function may change as it changes how it trades off errors across states. In our realizable setting, this changing importance may be less important, because our RQF predictor can perfectly represent the target. In our own experiments, we found the value bonuses did always converge to zero. Nonetheless, we know of no theory that would allow us to guarantee this.

**Connection to BDQN:** Though not obvious at first glance, there is a connection between RQFs and random prior functions in BDQN. In BDQN, the value function is $q_\theta = f_\theta + p$ for a random prior function $p$ that is not updated and learned function $f_\theta$. Random priors were developed for stationary state distributions—though then applied to control—so let us consider the update for a fixed policy $\pi$. The update uses $a' \sim \pi(\cdot|s')$, giving $r + \gamma(f_\theta(s', a') + p(s', a')) - (f_\theta(s, a) + p(s, a)) = r - (p(s, a) - \gamma p(s', a')) + \gamma f_\theta(s', a') - f_\theta(s, a)$. This is a standard update with reward bonus $p(s, a) - \gamma p(s', a')$, and this bonus is the negation of our reward in Equation (3). With a fixed policy, we can separate the value function learning into $q^\pi$ that estimates the values for the rewards and $b^\pi$ that estimates the values for the reward bonuses. Namely, $f_\theta$ consists of $q^\pi + b^\pi$. As these functions converge, $b^\pi(s, a)$ approaches $-p(s, a)$ using the exact same argument to the one in our Proposition 1, just negating the function $p$. Consequently, $f_\theta(s, a) + p(s, a) = q^\pi(s, a) + b^\pi(s, a) + p(s, a) = q^\pi(s, a) + (b^\pi(s, a) + p(s, a))$ goes to $q^\pi$ since $b^\pi(s, a) + p(s, a)$ eventually cancels.

This argument is not how randomized priors are presented, but provides another intuitive interpretation. Further, it highlights a key difference BDQN and VBE: BDQN takes a Thompson sampling approach to induce optimism whereas VBE acts greedily with respect to optimistic value estimates.

## 5 EXPERIMENTS

We evaluate our proposed algorithm on four classic exploration environments and six Atari environments, particularly in comparison to BDQN and the reward bonuses approaches ACB and RND. We first investigate the algorithms in a pure exploration setting, on DeepSea, where we evaluate state coverage. Then we compare performance on the classic environments, and investigate the impact of

the bonus scale and number of RQFs in the ensemble. We also compare variants of VBE which use ACB and RND's reward bonus strategy to to estimate the value bonuses (VB ACB and VB RND). We conclude with experiments in Atari, particularly highlighting how to scale VBE to this setting.

## 5.1 Environments

The four classic exploration environments are Sparse Mountain Car, Puddle World, River Swim and DeepSea. These four environments have varying requirements for exploration: DeepSea and River Swim are considered hard exploration environments, whereas Puddle World and Mountain Car require less exploration. The full details for these environments are in Appendix B, but we list a few key details here.

Mountain Car has two-dimensional continuous inputs, with a sparse reward structure: the agent only receives a reward of 1 at the goal and 0 otherwise. Puddle World also has two-dimensional continuous inputs, noisy actions and highly negative rewards in puddles along the way to the goal.

River Swim and DeepSea were both designed as hard exploration problems, requiring persistent behavior with likely failure under dithering, $\epsilon$-greedy exploration. River Swim resembles a problem where a fish tries to swim upriver, with high reward (+1) upstream which is difficult to reach and, a lower but still positive reward (+0.005), which is easily reachable downstream. This environment has a single continuous state dimension in $[0, 1]$, with stochastic displacement when taking actions left or right. One seemingly innocuous but important point for this environment is that we flipped the observation such that the high reward is at observation 0 and the lower reward is at observation 1. We did so because the standard random initialization and ReLU activation often results in a higher value for a higher input, thus favouring the correct action in this case. This other variant removes this inadvertent bias without changing the problem structure or difficulty in any way.

DeepSea is similar to River Swim, but is a grid world environment. Reaching the high-reward state requires the agent to take the action to go right every time. However, there is a penalty of $\frac{0.01}{N}$ for taking the action right, except for the right most state where the agent gets a reward of 1 for taking the right action. A policy that explores uniform randomly has the probability of $2^{-N}$ of reaching the goal state in each episode.

## 5.2 Algorithms and Experimental Settings

The environments uses slightly different evaluation metrics. River Swim is continuing, so we report accumulated reward over learning. For both Deepsea and Puddle World, we report the undiscounted episodic return. For Mountain Car, we report the discounted return, because for every successful episode, the undiscounted return is 1 and so not meaningful in this sparse variant. For all episodic environments, we report steps on the x-axis and the corresponding episodic return on y-axis. All results in the classic environments use 50000 steps and 30 runs, except DeepSea which uses 10000 episodes and 5 runs. The default grid size for Deepsea, unless mentioned, is 50, which is the largest grid size we experiment with.

Across problems we compare VBE with DQN with additive priors (DQN-P), BDQN, the released variants of ACB and RND that use PPO [3], and their DDQN-based variants: VB ACB and VB RND. DQN-P simply adds an additive prior to DQN, like BDQN has; it can be seen as BDQN with one value function in the ensemble. For both VBE and BDQN, we test using 1, 2, 8 and 20 value functions in the ensembles and bonus scales of 1, 3 and 10. ACB and RND also use the same bonus scales. To match their original implementations ACB uses an ensemble of 128 to estimate the reward bonus, and RND uses two deep neural network with multiple (64) nodes in the final layer as the target and predictor network for the reward bonus. All methods use the same neural network architectures, detailed in Appendix B.

We also include variants of VBE to provide evidence for the way we estimate the value bonuses with our ensemble errors. We include VBE-SL, meaning that instead of the TD update, we use a supervised learning update. We discussed in Section 3 that the errors for VBE-SL are likely to reduce too quickly, resulting in insufficient exploration; we test that hypothesis here. We also use the reward bonuses underlying ACB and RND to learn the value bonus and replace them with VBE's

---

[3]See https://github.com/JordanAsh/acb/tree/main

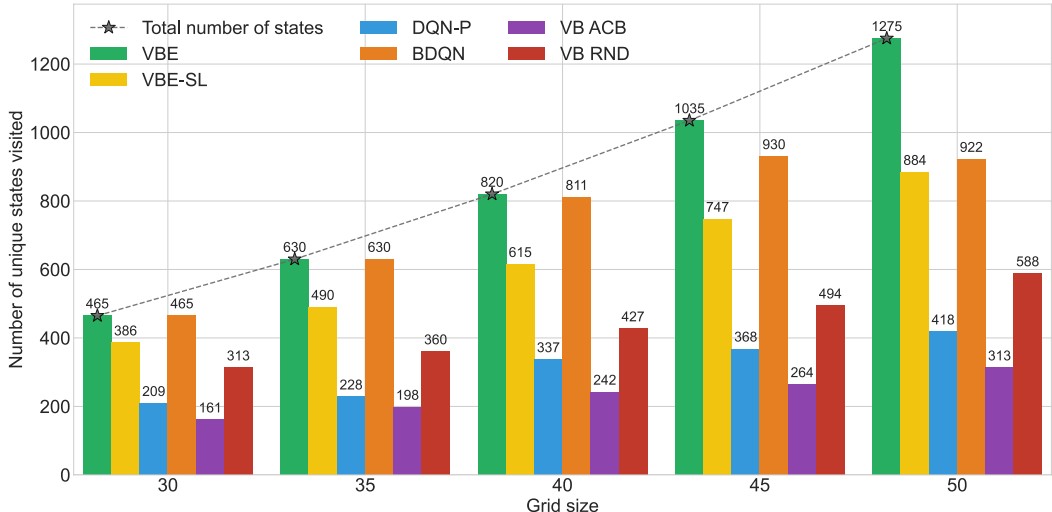

(a) State coverage in Deepsea of different grid sizes

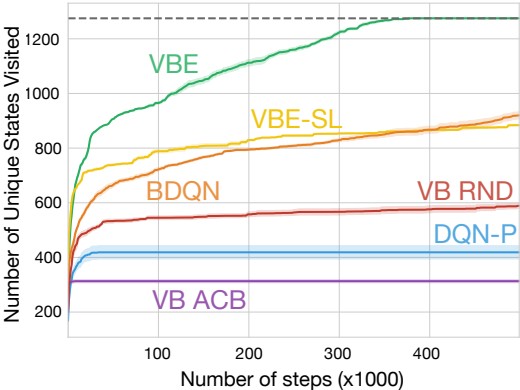

(b) Progression of unique states visited (grid size 50)

Figure 1: Contrasting the state coverage abilities of exploration algorithms in DeepSea. In (a) each bar corresponds to the total number of unique states visited by an agent after completing 10,000 episodes. The black stars indicate the total number of unique states visited for each grid size. Notably, VBE covers the entire state space, even for the larger grid sizes. (b) displays the progression of unique states visited by agents over the course of learning for Deepsea with grid size 50. The dotted line represents the total number of unique states (1275) in this environment. It provides evidence that VBE consistently explores new states at a significantly higher rate.

ensemble value bonus, i.e., VB ACB and VB RND. As proposed by the authors we make the reward-bonus value function non-episodic. VB ACB, VB RND and VBE-SL are otherwise exactly the same as VBE, including using DDQN, defining the value bonus using the ensemble error and using the value bonus in the same way.

### 5.3 PURE EXPLORATION

We first test how effectively the agents cover the state space in Tabular DeepSea with increasing grid sizes. For this tabular setting, the agents are otherwise the same as the other experiments, except the function approximation is linear on a one-hot encoding.

Figure 1a shows that VBE covers the entire state space for different grid sizes. BDQN is able to cover the state space for a grid size of 30 and 35, but starts to degrade after that. Both VB ACB and VB RND fail to cover the state space, with VB ACB covering even less than DQN-P. This outcome is not surprising, given that neither approach ensures first-visit optimism. VBE-SL at least includes first-visit optimism, encouraging the agent to take an action in a state if it has not done so before. But, as expected, it does not explore as much as VBE, likely as its value bonuses decay too quickly.

These suboptimal behaviors are emphasized in Figure 1b for a grid size of 50. All methods initially start exploring a similar number of states, easily reaching around 300 unique states. VB ACB, DQN-P and VB RND largely stop visiting new states very early in learning, though VB RND is slowly increasing the number of states it visits. BDQN and VBE-SL across in their behavior, with VBE-SL exploring more early, possibly due to better first-visit optimism. Over time, however, BDQN starts to catch up and then surpasses VBE-SL. VBE is the only algorithm that maintains a consistent increase until it has seen all states.

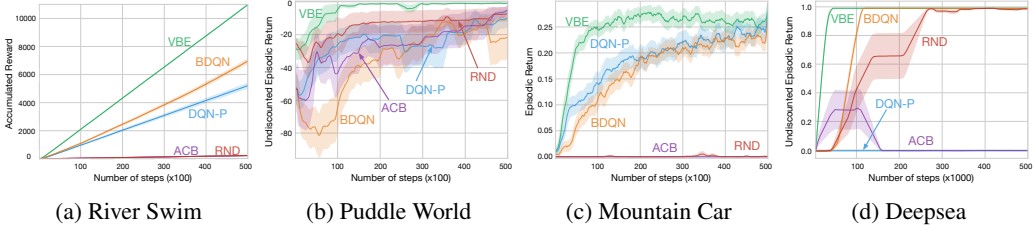

| (a) River Swim | (b) Puddle World | (c) Mountain Car | (d) Deepsea |

Figure 2: Comparison of online performance in River Swim, Puddle World, Mountain Car, and Deepsea. In all domains, higher on y-axis is better. The x-axis denotes the number of interaction steps with the environment. The shaded region corresponds to standard errors.

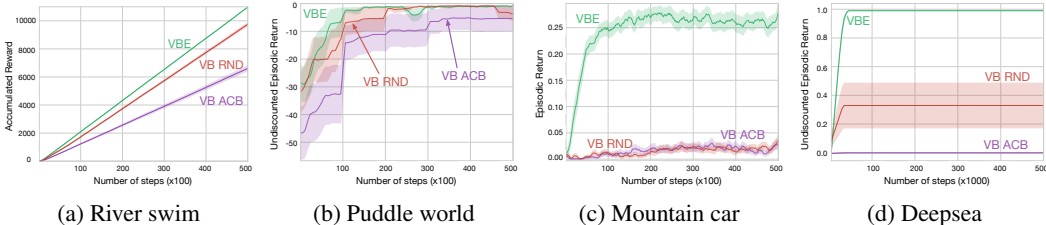

| (a) River swim | (b) Puddle world | (c) Mountain car | (d) Deepsea |

Figure 3: Comparing online performance VBE to two alternative ways to estimate value bonuses, namely estimating a value function on the reward bonuses given by RND and ACB. The shaded regions corresponds to standard errors.

## 5.4 COMPARISON IN CLASSIC ENVIRONMENTS

In this section we compare VBE with DQN-P, BDQN, ACB, and RND. ACB and RND here use PPO as originally proposed. In Figure 2, VBE learns faster and reaches the best final performance in all four environments. Surprisingly, DQN-P is competitive with BDQN in three out of the four environments. ACB and RND fail to learn in both the sparse reward domains Riverswim and Mountain Car. In Puddle World and Deepsea which have a denser reward structure, they perform better. RND learns slowly in both, whereas ACB is competitive in Puddle World, but does poorly in Deepsea.

## 5.5 ALTERNATIVE CHOICES FOR THE VALUE BONUS

We compared VBE to VB ACB and VB RND for pure exploration; now we do so for the four classic environments. VB ACB and VB RND are another natural way to estimate value bonuses—albeit missing first-visit optimism—and help validate our new approach to estimating value bonuses.

In Figure 3 we see that VBE outperforms VB ACB and VB RND. Similar to their PPO versions in Figure 2, they fail in Mountain Car and are competitive in Puddle World. However, behavior is quite different in Deepsea and Riverswim. Now VB ACB almost matches VBE in Riverswim, and VB RND has significantly improved compared to its PPO version. But, in Deepsea, they both perform notably more poorly, especially VB RND fails here but succeeded with its PPO version. The primary difference between them is using DDQN as a base algorithm, rather than PPO.

## 5.6 ATARI

In this section we test VBE on several hard exploration Atari games, namely Private Eye, Pitfall, Gravitar (Burda et al., 2019), and also on Breakout, Pong and Qbert. As is standard, we combine four consecutive frames to make the observation ($4 \times 84 \times 84$), and update VBE ever four steps. We do 3 runs for all the agents for 12 million steps.

In VBE the target and the predictor RQFs have 3 CNN-layers, followed by 2 non-linear layers (representation network) and a final linear output-layer. We only update the final layer of the predictor RQFs, and initialize the predictors to have the same representation network as the target RQFs, so to have learnable targets. In practice we found that even 1 RQF works well for VBE, so that is what we use in our experiments. We chose this configuration because it was faster to run and seemed to

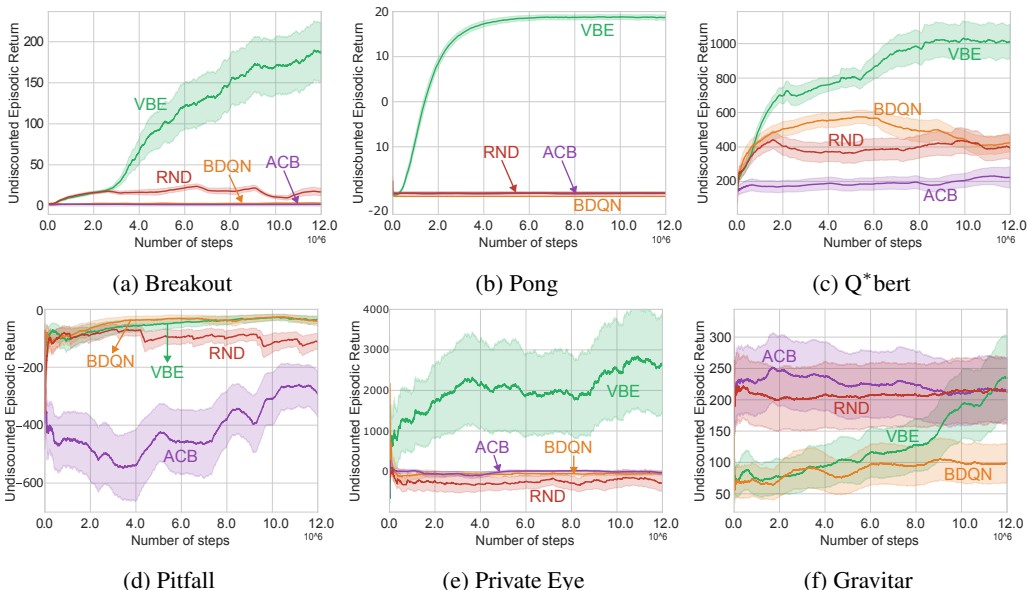

Figure 4: Comparing online performance in six Atari games, with shaded regions corresponding to standard errors. The environments in the second row are considered to be more challenging from an exploration perspective. The x-axis is the number of environment interaction steps in millions, and the y-axis is the online Undiscounted Episodic Return, for which higher is better.

be more stable than updating the whole network. We include the comparison to the variant of VBE where we update the whole network for the RQFs in Appendix D.

For BDQN we use an ensemble size of 10, and used a double-or-nothing bootstrap $p = 0.5$ (Owen & Eckles, 2012). As in the original BDQN implementation, each value function in the ensemble uses a shared representation network. The additive priors have the same network architecture as the value functions. ACB and RND agents do 128-step roll-outs and then do 4 epochs of network updates using PPO. To make sure that each agent gets the same number of interactions with the environment and to match our online setting, we run ACB and RND with one agent interacting with the environment instead of running multiple parallel agents. ACB uses an ensemble size of $k = 128$ for computing the reward bonus, and RND uses a CNN-based target and predictor. Note that VBE runs at least three times faster than BDQN, ACB and RND.

In Figure 4 we see that VBE outperforms the other algorithms in four out of six environments. This result is starkly different from the prior work, and is due to the fact that we examine early learning. RND was originally trained on around 2 billion frames, and ACB on around 20 million steps with data coming from 128 parallel agents. In Pitfall all algorithms except ACB are competitive with VBE. In Gravitar, RND and ACB perform surprisingly well from the beginning. VBE can learn to perform as well with more training steps, as can be seen in the result in Appendix Section D. Overall, these results show that VBE scales to more complex deep RL settings and results in sample efficiency improvements in early learning in several Atari environments. We also compare VBE with DDQN-based variants of ACB and RND in Appendix E.

## 6    CONCLUSION

In this work we introduced a new approach to do directed exploration in deep RL, called Value Bonuses with Ensemble errors (VBE). The utility of value bonuses is that it is simple to layer on top of an existing algorithm: the value bonuses are separately estimated and only impact the behavior policy. Improving how we estimate value bonuses, therefore, provides a promising path to replacing simple, but undirected exploration strategies like $\epsilon$-greedy. To date, the primary way to estimate value bonuses has been to estimate a separate value function on reward bonuses, as was done for ACB and RND. This approach, however, does not encourage first-visit optimism; it only encourages revisiting an action once a reward bonuses was observed. We show that, in general, ACB and RND do not provide effective exploration, in classic environments and several Atari environments, and that VBE consistently outperforms BDQN.

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
