# A  A DISCUSSION ON CONVERGENCE CRITERIA FOR VALUE BONUSES

First let us discuss how the theory for LSTD applies to our setting. The result from (Tagorti & Scherrer, 2015, Corollary 1) bounds the error of the value function learned under LSTD to the true value function, assuming features are linearly independent (Assumption 1) and a mixing assumption for the environment and behavior policy (Assumption 2). This bound includes an error to the best linear solution, for infinite data, and the error between the best linear solution and the true value function. Because we are in the realizable case and the objective is convex for linear function approximation, the best linear solution is the true value function and in the limit of data the LSTD solution will reach this best linear solution. We can write this as a corollary of their result. Note their result is written by value functions, but automatically extends to action-value function by considering state-action features and stationary distribution $\mu_b(s, a) = \mu(s)\pi_b(a|s)$.

**Corollary 1 (Corollary following from [Theorem 1)** *(Tagorti & Scherrer, 2015)] Assume we are given behavior policy $\pi_b$ with stationary distribution $\mu$ and target policy $\pi$ and the rewards are defined using a randomly sampled $f_i$ from the set of linear functions on features $\phi(s, a)$ and the formula in Equation* (3). *Under Assumption 1 and 2 from (Tagorti & Scherrer, 2015), for a large enough number of samples $T$ given by (Tagorti & Scherrer, 2015, Eq 6) (called $n$ in their result), then $f_{w_i}$ returned by LSTD satisfies*

$$\mathbb{E}_{s\sim\mu a\sim\pi_b(\cdot|s)}[(f_{w_i}(s, a) - f_i(s, a))^2] \leq O(1/\sqrt{T})$$

Now let us discuss how the work on neural TD applies to our setting (Cai et al., 2019). The result is proved for neural networks with a single hidden layer using a ReLU activation for the hidden layer, with the additional condition that the stationary distribution for the policy has a bounded density over states and the stepsizes decrease at a rate of $1/\sqrt{t}$. This result immediately implies that our $f_{w_i}$ should converge to $f_i$, because the global solution for this problem is $f_i$ because it is in the value function class. We state this as a corollary of their result here, to be clear about how it applies.

**Corollary 2 (Corollary following from [Theorem 4.6)** *(Cai et al., 2019)] Assume that 1) the policy $\pi$ is fixed with stationary distribution $\mu$, where $\mu(s)\pi(a|s)$ has bounded density across the space $x = (s, a)$ 2) the function class $\mathcal{F} = \{\frac{1}{\sqrt{m}}\sum_{j=1}^m b_j \max(x^\top w_j, 0)|W = (b_1, \ldots, b_m, w_1, \ldots, w_m), \|W - W(0)\|_2 \leq B\}$ for $x = (s, a)$, $W(0)$ a point at which the weights are initialized in the algorithm and $B$ some constant, 3) $\|x\|_2 = 1$ for all $x$ and the rewards are defined using a randomly sampled $f_i$ from $\mathcal{F}$ and the formula in Equation* (3), *and 4) the Neural TD algorithm (Algorithm 1 in (Cai et al., 2019)) is run for $T$ steps with stepsize $\eta = \min((1 - \gamma)/8, 1/\sqrt{T})$. Then the algorithm returns $f_{w_i}$ that satisfies*

$$\mathbb{E}_{W\sim,\mu\pi}[(f_{w_i}(s, a) - f_i(s, a))^2] \leq \frac{O(B^2)}{(1 - \gamma)^2\sqrt{T}} + O(B^2 m^{-1/2} + B^{5/2}m^{-1/4})$$

**Proof:** The result also requires that the reward magnitudes are all bounded, which they are by construction. Theorem 4.6 states that the outputted action-value function is bounded as above to the global optimum in the function class. Because $f_i(s, a)$ is in the function class, we know it is the global optimum. ∎

# B  EXPERIMENT DETAILS

## B.1  ENVIRONMENT DETAILS

**Mountain Car** is classic control problem of driving an underpowered car up a mountain. The original problem is set up as cost-to-goal, and here to frame it as a challenging exploration problem we offset the reward by 1, making it a sparse reward problem. The start state is sampled from the range $[-0.6, -0.4]$, which is the valley between two mountains, and the car starts with velocity zero.

**Puddle World** is a continuous state 2-dimensional world with $(x, y) \in [0, 1]^2$ with 2 intersecting puddles: (1) $[0.45, 0.4]$ to $[0.45, 0.8]$, and (2) $[0.1, 0.75]$ to $[0.45, 0.75]$. The puddles have a radius of 0.1 and the goal is the region $(x, y) \in [0.95, 1.0], [0.95, 1.0]$. The problem is cost-to-goal with

additional penalty for when the agent is either puddle. The penalty for being in a puddle is proportional to the distance of the agent from the center of the puddle, i.e., negative reward for being close to the center. The agent chooses a direction of movement, resulting in displacement equal to $0.005 + \zeta, \zeta \sim N(\mu = 0, \sigma = 0.1)$ in the chosen direction. The starting positions for episodes is uniformly sampled from $(x, y) \in [0.1, 0.3], [0.45, 0.65]$. High variance transitions coupled with high magnitude penalties make this a challenging exploration problem.

**River Swim** is a standard continuing exploration benchmark inspired by a fish trying to swim upriver, with high reward (+1) upstream which is difficult to reach and, a lower but still positive reward (+0.005), which is easily reachable downstream. The state space is continuous in $[0, 1]$, and the stochastic displacement is equal to $0.1 + \zeta, \zeta \sim N(\mu = 0, \sigma = 0.01)$ in the direction of the chosen action up or down. As swimming upstream is difficult, action up is stochastically switched to down. We also flip the observation such that the high reward is at observation $0$ and the lower reward is at observation $1$. We do this because we noticed that using random initialization with RelU activations would mostly result in a higher value for a higher input thus favouring the correct action in this case. The starting position is sampled uniformly in $[0.9, 1.0]$.

**DeepSea** is a hard exploration episodic grid world environment. In each state the agent can take two actions, left or right, which moves the agent down one row with column shifting based on left or right action. Collisions to the grid edges are handled by the agent staying in the same column but moving down one row. Since the agent can never access the states on the right side of the diagonal of the grid, the total number of states are thus $\frac{N \times (N+1)}{2}$. The most rewarding state is the state on the bottom right corner of the grid. To reach this the agent to take the action to go right every time. However, there is a penalty of $\frac{0.01}{N}$ for taking the action right, except for in this high rewarding state where the agent gets a reward of 1 for taking the right action. This makes it a very challenging environment. A policy that explores uniform randomly has the probability of $2^{-N}$ of reaching the goal state in each episode.

## B.2 Algorithm Details

In the classic environments, every agent uses the same neural architecture, containing 2 non-linear layers with 50 nodes each and ReLU activation, followed by a linear output-layer. DQN-P, BDQN and all variants of VBE use target networks which are updated periodically after every $\tau$ steps. For DQN-P and BDQN we use $\tau = 4$ for all four classic environments. VBE and its variants use $\tau = 4$ for Mountain Car, Puddle World and River Swim, and $\tau = 64$ for DeepSea. We use a learning rate of $\alpha = 0.001$ and a discount factor of $\gamma = 0.99$. DQN-P, BDQN and VBE variants use an experience replay buffer that stores the most recent 50K transitions. The agent's parameters are updated after every step using a randomly sampled mini-batch of $128$. We sweep the agents on bonus scales $c = [1.0, 3.0, 10.0]$, and ensemble sizes $k = [1, 2, 8, 20]$. The PPO version of ACB uses an ensemble size of $k = 128$, and RND uses a multi-layer neural network instead of an ensemble. Tables 1, 2, 3 show the best performing sets of ensemble size $k$ and bonus scale $c$ for results in Sections 5.3, 5.4, 5.5.

|  | DeepSea |
| --- | --- |
| VBE | $k = 1, c = 1.0$ |
| VBE-SL | $k = 20, c = 1.0$ |
| DQN-P | $k = 1, c = 1.0$ |
| BDQN | $k = 20, c = 1.0$ |
| VB ACB | $k = 20, c = 1.0$ |
| VB RND | $c = 1.0$ |

Table 1: Ensemble size $k$ and bonus scale $c$ for agents in Figure 1

## C Linear function approximation

In this section we test VBE and the baseline agents on the same four classic environments as in Section 5.4, with tile-coded features and a single linear-layer network. For Riverswim we use the following tile-coding parameters: $(tiles = 4, tiling = 32, features = 128)$, Puddle world:

|  | River Swim | Puddle World | Mountain Car | DeepSea |
|---|---|---|---|---|
| VBE | $k = 20, c = 1.0$ | $k = 1, c = 10.0$ | $k = 2, c = 1.0$ | $k = 20, c = 1.0$ |
| DQN-P | $k = 1, c = 10.0$ | $k = 1, c = 3.0$ | $k = 1, c = 1.0$ | $k = 1, c = 10.0$ |
| BDQN | $k = 8, c = 10.0$ | $k = 2, c = 1.0$ | $k = 20, c = 1.0$ | $k = 20, c = 10.0$ |
| ACB ($k = 128$) | $c = 1.0$ | $c = 3.0$ | $c = 1.0$ | $c = 10.0$ |
| RND | $c = 1.0$ | $c = 10.0$ | $c = 10.0$ | $c = 1.0$ |

Table 2: Ensemble size $k$ and bonus scale $c$ for agents in Figure 2

|  | River Swim | Puddle World | Mountain Car | DeepSea |
|---|---|---|---|---|
| VBE | $k = 20, c = 1.0$ | $k = 1, c = 10.0$ | $k = 2, c = 1.0$ | $k = 20, c = 1.0$ |
| VB ACB | $k = 20, c = 10.0$ | $k = 1, c = 1.0$ | $k = 8, c = 1.0$ | $k = 20, c = 1.0$ |
| VB RND | $c = 10.0$ | $c = 1.0$ | $c = 1.0$ | $c = 1.0$ |

Table 3: Ensemble size $k$ and bonus scale $c$ for agents in Figure 3

($tiles = 5, tiling = 5, features = 128$), and Mountain car: ($tiles = 4, tiling = 16, features = 512$). The results in Figure 5 are similar to their neural network counterpart results in Figure 2. In Riverswim, RND does well and even surpasses BDQN in terms of performance. ACB, however, still fails on Riverswim. In Puddle world RND is comparable towards the end of training, and ACB is much slower. DQN-P outperforms BDQN in Puddle world, and Mountain car, whereas both ACB and RND fail in Mountain car. In Deepsea we see that DQN-P and ACB fail to learn the optimal policy. RND learns relatively quickly but then fails to stick to the optimal policy and thus collects less reward per episode throughout. In Figure 6 we compare VBE with VB ACB and RND in the linear setting. In Riverswim VB ACB does better than its PPO counter part in Figure 5a. VB ACB and RND perform comparable to VBE in Puddle world and Mountain car. Both VB ACB and RND fail in Deepsea.

## D   ATARI

In this section we compare two variants of VBE on the same six Atari games. VBE as mentioned in Section 5.6 only updates the final output layer of predictor RQFs. We implement a version of VBE, VBE-CNN which updates the complete CNN architecture for predictor RQFs. Since we are updating the complete network the predictor and target RQFs do not need same weights for the representation network. To ensure that the magnitude of errors is not too small, we initialize the target RQFs with a scale of 1 (scale parameter in Orthogonal initialization), and initialize predictor RQFs with a scale of 0.01. Note that this choice of scale was adopted from ACB's public implementation by Ash et al. (2022). We use an ensemble size of $k = 1$ for both these agents, and a bonus scale of $c = 10$ for all environments, except for Pitfall for which we use a bonus scale of $c = 1$ for all the agents. In Figure 7 we see that both agents continue to learn and improve at around 12 Million frames. VBE is better than VBE-CNN in Breakout, Pong and Privateye. However, in Pitfall VBE-CNN converges to a policy that results in no negative reward at around 6 Million frames. This result is quite impressive as Pitfall is a very hard exploration environment and most agents take a lot more data to learn a good policy in this environment. VBE does exceptionally well on Private Eye, which is also a very hard exploration environment. In Q*bert we see that both agents are comparable uptill 6 Million frames after which VBE-CNN first decreases in performance but eventually surpasses VBE towards the end. Finally, in Gravitar we see that both agents are competitive and continue to improve. Both variants of VBE do well in all six Atari environments, and demonstrate sample efficiency by early learning. Although we do not run the agents for as long as the baselines, VBE is still able to learn much quickly and in some environments even outperforms baselines trained on much more data.

## E   ATARI: ALTERNATIVE CHOICES FOR THE VALUE BONUS

In this section we compare VBE with DDQN-based variants of ACB and RND, denoted as VB ACB and VB RND, on Atari to see early learning. In Figure-8 we can see that the VB ACB and VB RND are much more sample-efficient than their PPO-based counterparts. However, VBE still performs better in general. VBE performs the best in three out of six environments, i.e., in Breakout, Pong,

|            | VBE       | VBE-CNN     |
|------------|-----------|-------------|
| Breakout   | **170.17**| 73.08       |
| Pong       | **18.81** | 5.65        |
| Qbert      | 963.73    | **2026.31** |
| Pitfall    | -18.65    | **0.0**     |
| Privateeye | **2416.05**| 242.12     |
| Gravitar   | **247.76**| 214.6       |

Table 4: Mean episodic return observed during the last 500 episodes of online training.

Privateeye. In Qbert, VBE is better than VB ACB, and in Pitfall VBE is better than VB RND. In Gravitar both VB ACB and VB RND perform better than VBE in the given frame budget, but we know from Figure-7 that VBE is slow early on but continues to improve beyond 5 million frames. Also, in Pitfall we note that beyond 5 million frames VBE surpasses VB ACB. We will update the paper with results on more frames in the final version.

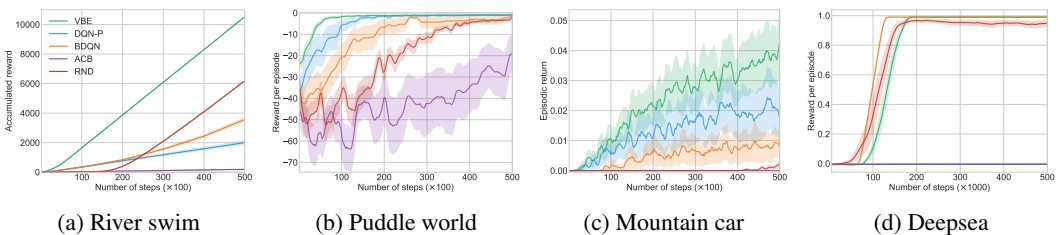

(a) River swim  (b) Puddle world  (c) Mountain car  (d) Deepsea

Figure 5: Comparing VBE with baseline agents in four classic control environments, with tile-coded features and a linear-layer.

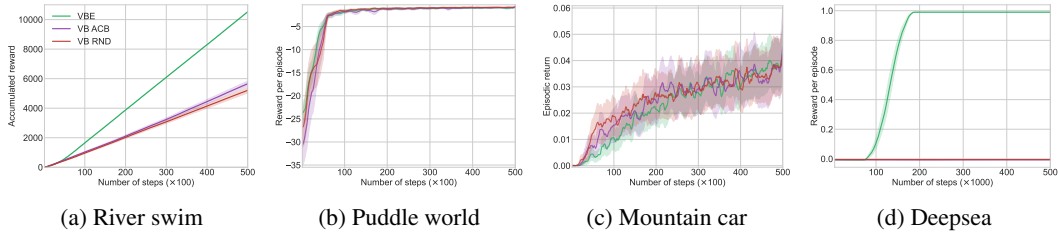

(a) River swim  (b) Puddle world  (c) Mountain car  (d) Deepsea

Figure 6: Comparing VBE with VB ACB and VB RND in the linear setting with tile-coded features.

## F    ENSEMBLE SIZE × BONUS SCALE

In this section we show the effect that bonus scales and ensemble sizes have on the performance of the VBE in each of the four classic control environments. In Figure 9 we show the average performance of VBE used in Section 5.4, for each environment. For Riverswim we see that the performance improves as the bonus scale and the ensemble size is increased. This makes sense as Riverswim is a hard exploration environment and requires more aggressive exploration. In Puddle world and Mountain car, we observe that increasing the bonus scale and the ensemble size harms the performance, since they do not rquire too much exploration. For Deepsea we only test a bonus scale of 1 with different ensemble sizes on different grid sizes. We can see that only an ensemble size of 20 works well on all grid sizes. For the single linear-layer agent we observe a similar patter in Figure 10.

## G    TARGET POLICY EXPERIMENTS

In many cases it is straight forward to define the agent's behaviour policy depending on the problem at hand, for example, for exploration an agent can use an $\epsilon-$greedy policy, or use an upper

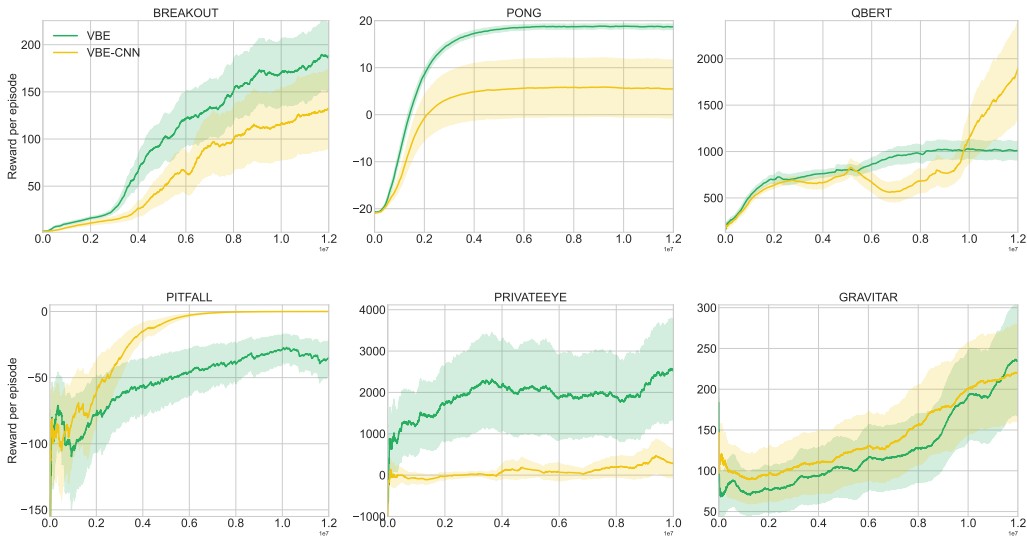

Figure 7: Comparing the online performance of VBE and VBE-CNN on six Atari environments. For Private Eye we compare the agents for 10 Million steps, and the rest for 12 Million steps.

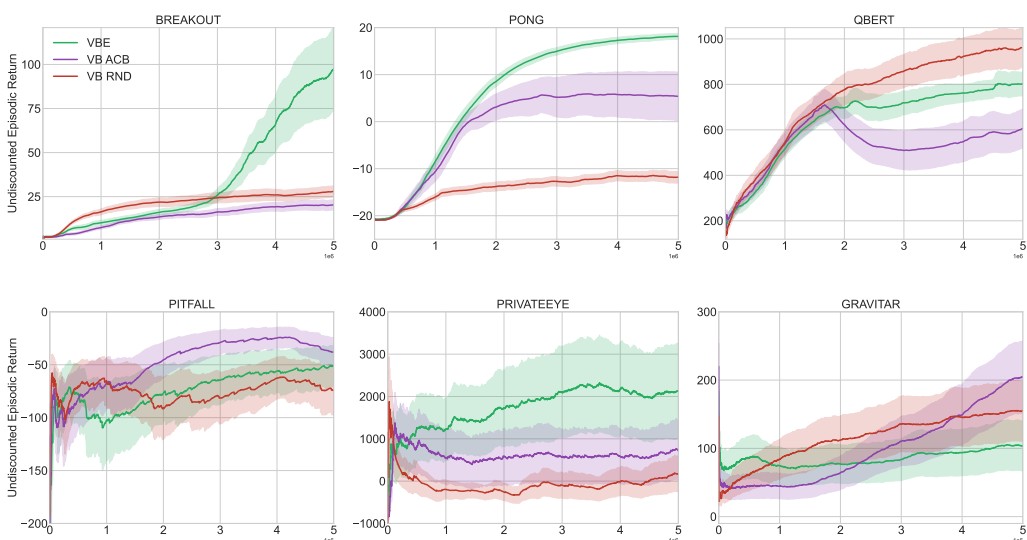

Figure 8: Comparing the online performance of VBE, VB ACB and VB RND on six Atari environments on a budget of 5 million steps to see early learning.

confidence style bonus to select actions, lets call this the optimistic policy. However, in case of the general policy iteration setting it is not clear what the target policy should be to perform updates. Should the target policy also be optimistic/on-policy? Or should it be greedy only with respect to the value estimates/off-policy? The question becomes especially challenging to answer when there are multiple value functions or an ensemble of value functions, like in our case. What should be the target policy to update the RQFs? To investigate we performed a series of experiments using different types of target policies, i.e., optimistic, and greedy. The optimistic agent uses the optimistic policy to update the value head and the RQFs. The greedy agent uses the greedy target policy to update both the value head and the RQFs. The target policy for the value heads and the RQFs is consistent, this is because we want the RQFs to correspond to the value estimates. The intuition behind using an optimistic policy is that the target action-values it gives may have high prediction errors and up-

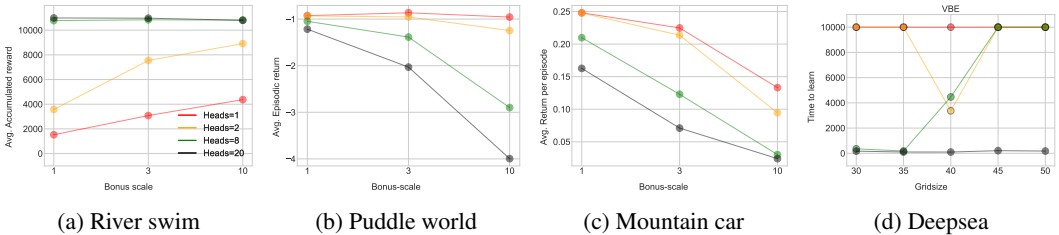

|  |  |  |  |
| --- | --- | --- | --- |
| (a) River swim | (b) Puddle world | (c) Mountain car | (d) Deepsea |

Figure 9: Shows the effect of different bonus scales and ensemble sizes across the classic control environments. For Deepsea, we only use a bonus scale of 1 and test different ensemble sizes.

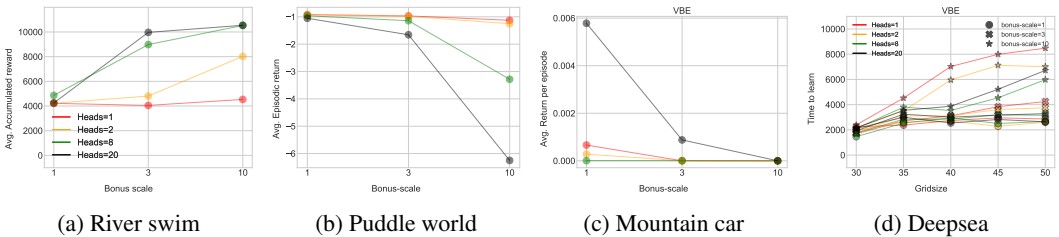

|  |  |  |  |
| --- | --- | --- | --- |
| (a) River swim | (b) Puddle world | (c) Mountain car | (d) Deepsea |

Figure 10: Shows the effect of different bonus scales and ensemble sizes across the classic control environments. These results correspond to the single linear-layer agent.

dating the RQFs by bootstrapping off of values with high prediction error is likely to produce errors in current RQF predictions as well. This allows uncertainty to propagates allowing the agent to do directed exploration. In case of the greedy target policy, the uncertainty still propagates however, in this case the prediction errors don't play a role in the selection of action-values, for example, an action-value with a low value but a high prediction error may not be selected using a greedy target policy.

In Figure 11 we show two different agents, optimistic and greedy, run on different grid sizes of DeepSea with different ensemble size, represented by color, and different bonus scales, represented by shapes. The agents use a single linear-layer for the value function and RQFs. We can see that the greedy agent generally performs better than the optimistic agent, which makes sense as the optimistic target policy can cause more exploration. The greedy agent performs well even with multiple RQFs, whereas optimistic agent fails to learn the optimal policy as the number of RQFs increase. We use the greedy target policy version of VBE in all the results of the main body.

In our experiments with VB with ACB and RND, we noticed a strange phenomenon, i.e., using an optimistic target policy allows VB ACB and VB RND to learn the optimal policy quickly on DeepsSa environments (Figure 12), and using a greedy target policy for VB ACB/RND would cause the agents fail to learn the optimal policy (Figure 6d, 3d). This is interesting, as reward bonus methods do not provide optimism for unseen action-values, VB with ACB and RND should not be able to cover the entire state space based on random initialization. We found out that this happens because of the bias term in the linear layer, the momentum term in the optimizer and because the intrinsic value function is non-episodic. Using the optimistic target policy and with the help of momentum the bias term consistently increases, consequently the intrinsic action values start to increase. Since the bias-term is a shared parameter, the increase in its value provides the optimism for unseen action-values as well, this allows for the agent to cover the state space and thus learn the optimal policy. In case of the greedy target policy the intrinsic values do no increase, thus the agent fails to cover the state space. In Section 5.3, we show that if we use tabular features and a linear-layer without any bias term then VB ACB/RND fail to cover the state space. In this setting the agent's behaviour and target policy is governed by only the intrinsic value functions (on-policy), however it fails on account of not providing first-visit optimism.

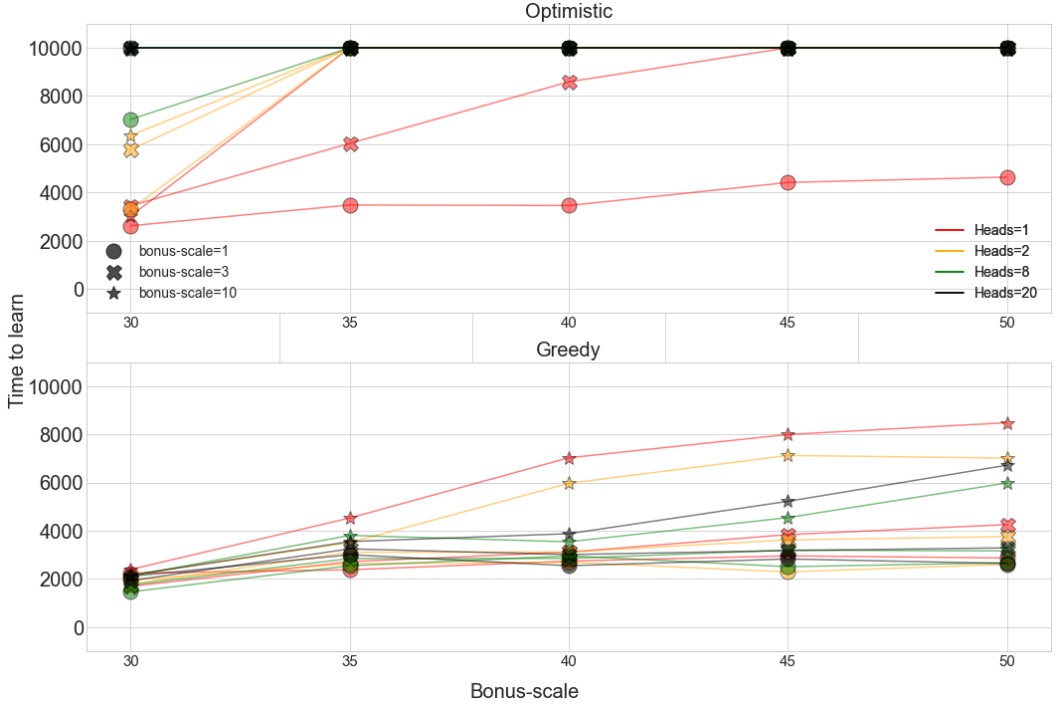

Figure 11: Comparing Optimistic target policy (On-policy) with Greedy target policy (Off-policy) on different grid sizes of Deepsea.

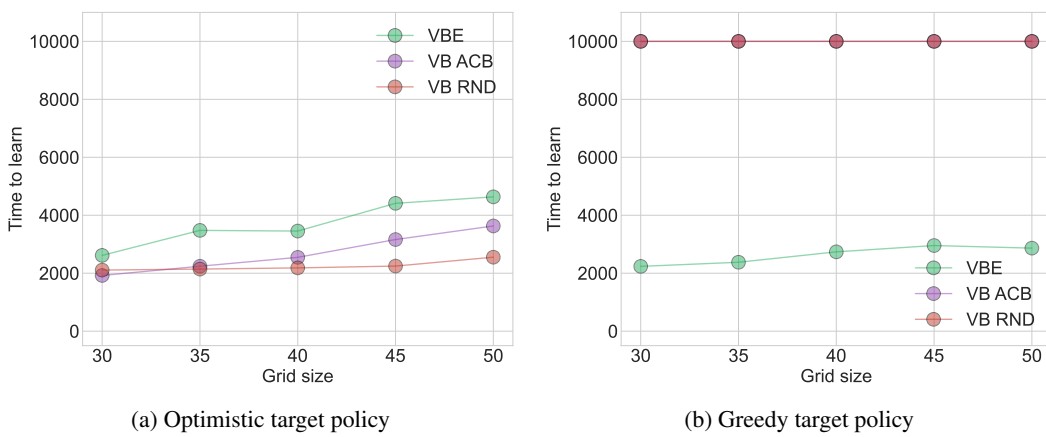

(a) Optimistic target policy

(b) Greedy target policy

Figure 12: In Figure 12a the agents do on-policy (optimistic) updates. In Figure 12b the agents do off-policy (greedy) updates. VB ACB/RND fail with the greedy policy, whereas with optimistic target policy they outperform VBE. VBE however, does well with a greedy policy compared to the optimistic one. These agents use a single linear-layer with bias term.