# OpenReview forum: "Value Bonuses using Ensemble Errors for Exploration in Reinforcement Learning"
_ICLR.cc/2024/Conference — Submitted to ICLR 2024_

### Official Review · Reviewer_E6HL · 2023-10-30

**Soundness:** 2 fair
**Presentation:** 3 good
**Contribution:** 2 fair
**Rating:** 5
**Confidence:** 4

**Summary:**

This paper introduces an approach to improving the exploration capabilities of value-based reinforcement learning (RL) algorithms by incorporating a value bonus. Unlike previous studies that define exploration bonuses as reward distributions, this method offers optimistic exploratory value even for unvisited state-action pairs. The technique involves maintaining an ensemble of random Q-functions and predictors that estimate the underlying reward. The behavior policy then selects greedy actions based on the augmented Q-function.

The experimental results demonstrate the effectiveness of this bonus when applied to the Double Deep Q-Network (DDQN) algorithm in both classical environments and some games in the Atari suite. The agent optimizes state coverage, leading to improved rewards. Overall, the paper presents a promising approach to enabling deep exploration in RL algorithms.

**Strengths:**

The paper proposes a method to facilitate deep exploration in RL by combining an ensemble of Q-functions and random target functions. This method, which builds on prior research in uncertainty estimation and exploration strategies, offers a straightforward implementation of DQN variants and enhances exploration more effectively than traditional strategies like epsilon-greedy. The paper highlights a key strength: the ability to allocate optimistic value estimates to unvisited state-action pairs, a feature that methods that rely solely on reward bonuses do not offer.

While the experiments are somewhat limited in terms of computational requirements, they are designed to support the paper's claims. VBE achieves promising results in some interesting classical experiments. These experiments show that VBE can potentially replace undirected exploration strategies in value-based RL algorithms. However, the overall narrative in the experiments section becomes unclear due to the non-standard experimental settings. The comparisons between VBE and the baselines are further complicated by the differences in RL algorithms (DDQN vs. PPO) and allowed environment steps, where DDQN's superior sample efficiency makes the comparisons unfair.

In summary, the paper presents a promising method for deep exploration in RL, although further clarification and additional experiments comparing different RL algorithms could strengthen the paper's overall argument.

**Weaknesses:**

The initial sections of the paper are well presented, providing a clear introduction to the research. However, a more comprehensive contextualization with existing prior work on deep exploration, specifically related to value bonuses, would enhance the paper's relevance. A thorough literature review would help readers understand the specific problem and knowledge gap this work addresses.

The experiments section, while exploring interesting environments, falls short of the standards expected for empirical studies in deep RL. While the selected environments are interesting for validating the proposed method, a more impactful study would involve challenging and widely recognized hard-exploration tasks. Additionally, the experimental settings do not align with the standard parameters for fair evaluation of deep RL algorithms, particularly in the context of hard-exploration tasks. The comparison between different RL algorithms (DDQN, PPO) under such limited environment interaction raises concerns about the fairness of the evaluation. It remains unclear whether the observed learning curve superiority of VBE after 50k steps is due to DDQN's known sample efficiency compared to PPO.

In summary, the paper introduces an interesting idea and presents it effectively. However, the current experiments, while promising, lack the necessary impact due to the limitations in experimental design and evaluation. Addressing these concerns and conducting experiments on more challenging and standard hard-exploration tasks would significantly enhance the paper's significance.

**Questions:**

I am curious about the author's choice to allocate computational resources to run 30 different experiments, a number significantly higher than the typical 5-10 runs in deep RL. Instead, I would suggest using some of these resources for longer experiments, which might have offered a more fair analysis given the well-known sample inefficiency of deep RL algorithms.

I appreciate the authors' effort in providing the reference codebase for implementing the RND and ACB baselines. However, I am not convinced about the effectiveness of this particular implementation, given the limited popularity of the codebase. I am concerned that this implementation might not be able to replicate the results of the proposed baselines, especially for RND. I would suggest using the official implementations or more widely validated ones.

Additionally, I wonder if the authors had considered exploring other environments that offer challenging exploration tasks. Environments involving 3D navigation or procedurally generated maps could potentially provide a more impactful evaluation of the proposed method. Including such environments in the experimental setup could strengthen the paper's contribution.

---

> ### Author Response · Authors · 2023-11-16
>
> “a more impactful study would involve challenging and widely recognized hard-exploration tasks”
>
> The environments we chose in our experiments was to make sure that we have a balanced mix of hard exploration environments and relatively simpler environments. The reason is because we wanted to present an algorithm that does well in general rather than just on hard exploration problems. In this context, it is note-worthy that existing literature highlights that focusing only on hard-exploration problems while evaluating exploration algorithms can lead to a very skewed view of their general efficacy [14].
>
> The environments we chose are pretty standard to test exploration capabilities of RL agents and have previously been used in many studies. Riverswim [1, 2, 3, 4] with a continuous state-space and noisy transitions is a continuing environment. It is a hard exploration environment as the agent starts near the sub-optimal reward and the probability of getting to the optimal reward state is low. Puddle-World [1] tests the agent’s ability to effectively explore the environment even in the presence of high magnitude negative reward for stepping inside the puddles. The agent not only has to find the shortest path to the goal but also has to learn to avoid going inside the puddles. Sparse Mountain Car [1, 3] is a variant of the classic Mountain Car environment, and highlights the difficulty of learning a good policy in absence of a dense reward. DeepSea [3,5,6,7,8] is widely regarded as a hard exploration environment as there exists only one correct path towards the goal state and throughout this path the agent receives a penalty up till the very end. A random exploration strategy would need exponential samples O(2^-N) to reach the goal state. For Atari we also chose a mix set of hard exploration environments like: Gravitar, Pitfall, Privateeye [9,10,11], and the remaining Breakout, Pong and Qbert [11, 12] were selected to test the ability of VBE and baseline agents on relatively simpler environments.
>
> “The comparison between different RL algorithms (DDQN, PPO) under such limited environment interaction raises concerns about the fairness of the evaluation.  It remains unclear whether the observed learning curve superiority of VBE after 50k steps is due to DDQN's known sample efficiency compared to PPO. ”
>
> As mentioned below, we used the code released for ACB/RND, which uses PPO in Atari, precisely because we wanted to fairly compare to an implementation that was well-developed by those authors. We do, however, completely agree that the idea is separable from PPO and can and should be tested within DDQN. This is why we did just that in the more controlled experiments (rather than Atari, where we wanted to compare to known baselines). In Section 5.5 we evaluate ACB and RND with DDQN instead of PPO. We use the exact same base algorithm as used for VBE to test ACB and RND’s bonus. We also test these DDQN variants of ACB and RND in the pure exploration experiment, and highlight that even with DDQN implementation these reward bonus strategies fail to cover the state-space because of their limitation to provide first-visit optimism.
>
> “I appreciate the authors' effort in providing the reference codebase for implementing the RND and ACB baselines. However, I am not convinced about the effectiveness of this particular implementation, given the limited popularity of the codebase.”
>
> The codebase we used for ACB and RND is the one written by the authors of the ACB paper [11]. The codebase is built on top of [13] which is a much more popular and widely used implementation of RND. We also made sure that the implementation was sound and used the parameters originally proposed.
>
> “I am curious about the author's choice to allocate computational resources to run 30 different experiments”
>
> Our goal in exploration is to improve sample efficiency, as directed exploration should result in the agent learning with fewer samples. It is sensible, therefore, to examine early learning to gauge this sample efficiency. Testing exploration algorithms under a fixed budget is also a common experiment setup [1, 3, 5, 7, 8]. About computation, when using clusters to run experiments, it is straightforward to schedule more jobs to get more runs, since they run in parallel. Each job runs in a more reasonable amount of time than a much longer experiment, and allows for iteration on the results to better understand the methods. Also, doing 30 runs with random seeds lead to more statistically significant results.
>
> “Environments involving 3D navigation or procedurally generated maps ”
>
> As stated above the environments we chose in our experiments are pretty standard to test exploration capabilities of RL agents. However, this is a nice suggestion and can be considered for future work.
>
> We hope these responses resolve your concerns, and would be happy to hear any other specific issues with the experimental setup. Thank you for your review and input.

---

> > ### Author Response · Authors · 2023-11-16
> >
> > References:
> >
> > [1] Kumaraswamy, R., Schlegel, M., White, A., & White, M. (2018). Context-dependent upper-confidence bounds for directed exploration. Advances in Neural Information Processing Systems, 31.
> >
> > [2] Osband, I., Russo, D., & Van Roy, B. (2013). (More) efficient reinforcement learning via posterior sampling. Advances in Neural Information Processing Systems, 26.
> >
> > [3] Ishfaq, H., Cui, Q., Nguyen, V., Ayoub, A., Yang, Z., Wang, Z., ... & Yang, L. (2021, July). Randomized exploration in reinforcement learning with general value function approximation. In International Conference on Machine Learning (pp. 4607-4616). PMLR.
> >
> > [4] Strehl, A. L., & Littman, M. L. (2008). An analysis of model-based interval estimation for Markov decision processes. Journal of Computer and System Sciences, 74(8), 1309-1331.
> >
> > [5] Osband, I., Doron, Y., Hessel, M., Aslanides, J., Sezener, E., Saraiva, A., ... & Van Hasselt, H. (2019). Behaviour suite for reinforcement learning. arXiv preprint arXiv:1908.03568.
> >
> > [6] I Osband, B Van Roy, and Z Wen. Generalization and exploration via randomized value functions. In International Conference on Machine Learning, 2016.
> >
> > [7] Ian Osband, John Aslanides, and Albin Cassirer. Randomized Prior Functions for Deep Reinforcement Learning. NeurIPS, 2018.
> >
> > [8] Ian Osband, Benjamin Van Roy, Daniel J Russo, Zheng Wen, et al. Deep exploration via randomized value functions. J. Mach. Learn. Res., 20(124):1–62, 2019.
> >
> > [9] Bellemare, M., Srinivasan, S., Ostrovski, G., Schaul, T., Saxton, D., & Munos, R. (2016). Unifying count-based exploration and intrinsic motivation. Advances in neural information processing systems, 29.
> >
> > [10] Yuri Burda, Harrison Edwards, Amos Storkey, and Oleg Klimov. Exploration by random network distillation. In International Conference on Learning Representations, 2019. URL https:
> > //openreview.net/forum?id=H1lJJnR5Ym.
> >
> > [11] Jordan T. Ash, Cyril Zhang, Surbhi Goel, Akshay Krishnamurthy, and Sham M. Kakade.
> > Anti-concentrated confidence bonuses for scalable exploration. In International Conference on Learning Representations, 2022. URL https://openreview.net/forum?id=
> > RXQ-FPbQYVn.
> >
> > [12] Osband, I., Blundell, C., Pritzel, A., & Van Roy, B. (2016). Deep exploration via bootstrapped DQN. Advances in neural information processing systems, 29.
> >
> > [13] https://github.com/jcwleo/random-network-distillation-pytorch
> >
> > [14] Taiga, A. A., Fedus, W., Machado, M. C., Courville, A., & Bellemare, M. G. (2021). On bonus-based exploration methods in the arcade learning environment. arXiv preprint arXiv:2109.11052.

---

> > ### Comment · Reviewer_E6HL · 2023-11-16
> >
> > Thank you for addressing my points. While I still believe that the experiments section is not of the same quality as the presentation of other sections in the paper, I am now more convinced about the usefulness of the knowledge that can be learned from the current experiments. I still think that the results do not meet the requirements for supporting a publication at ICLR, but I slightly increase the score of my evaluation as I acknowledge that the current evaluation of VBE shows that it is indeed a promising method.

---

> > > ### Author Response · Authors · 2023-11-23
> > >
> > > Thank you for your response! We have updated the paper and have addressed the concerns raised by the reviewer. Primarily, we have now added a new result in the paper (Appendix E ; Figure-8), where we compare VBE with DDQN-based variants of ACB and RND, namely VB ACB, VB RND. Previously we only compared VB ACB/RND in the pure exploration experiment and in the four classic control environments. The reviewer was right in noting that DDQN-based variants of ACB and RND are more sample-efficient than their PPO-based counterparts. However, we can see in Figure-8 that VBE still performs better in general, outperforming in 3/6 environments. In Qbert VBE performs better than VB ACB, and in Pitfall VBE performs better than VB RND. We also note that beyond 5 million frames on Pitfall, VBE surpasses VB ACB too (will update the paper). It is only in Gravitar where both VB ACB/RND are better than VBE, as VBE is slow but continues to improve beyond 5 million frames. We will update the paper with results on more frames in the final version.
> > >
> > > We hope that this new result further satisfies the reviewers concerns about fairness and the experimental setup. Thank you!

---

### Official Review · Reviewer_pttX · 2023-10-31

**Soundness:** 4 excellent
**Presentation:** 3 good
**Contribution:** 3 good
**Rating:** 6
**Confidence:** 3

**Summary:**

Current SOTA exploration bonuses are only retroactive: that is they reward states *infrequently* visited, and then rely on randomness around those infrequent states to constantly expand the horizon of exploration. This paper proposes an exploration bonus that rewards states *never* visited through uncertainty in an ensemble of Q-functions. The ensemble is trained to approximate the next-state of a random target function, not the current state like RND, and during rollout, the behavior is taken as a max over actions for the current value for the problem reward and the bonus derived from the ensemble disagreement. There is some discussion of the theory, and some small-scale and mid-scale experiments to support the method, with strong results.

**Strengths:**

- Exploration bonuses seem underexplored as of late, especially given that RND suffers from only rewarding infrequent states and can collapse like the authors show (I have personal experience with this as well), if we are interested in settings where behavior data is not available, we will need better exploration methods
- The method seems quite sample efficient in empirical evaluations, and the authors correctly note that RND takes an extremely large number of samples to converge (2 billion in the original paper), which is very undesirable if we want to move out of simulation
- Discussion surrounding related literature and motivation of the argument is very sound

**Weaknesses:**

- The experiments are a bit small-scale (only ~400k environment steps at most in the Atari domains)
- There is no experiment in the main text on the larger domains that runs to completion, only the early exploration behavior, while I agree that early exploration behavior is more informative for our understanding, it would be good to have an example of how more frames changes behavior in more difficult settings (currently in the Appendix Figure 7, but doesn't show baselines as well)
- There is little discussion on the relatively worse performance in Pitfall and Gravitar in Figure 4, I believe some closer qualitative analysis is merited
- It would be nice to have a showcase result on Montezuma's revenge, or perhaps one of the Antmaze tasks for a more difficult sparse-reward setting?
- There is no explicit objective given for q_w in Algorithm 1, even though the reward has been relabeled in line 12, when we aren't in the pure exploration setting are we using the original reward? I think it would be worthwhile in the main text to give the exact objectives for all components in one place
- It's possible that the really bad failures of RND in Mountain Car are due to the simplicity of the state space, which leads to an early collapse in the bonus, it would be nice to have some discussion of why this occurs in Figure 3.

Minor Comments:
- Section 3, the paragraph "We need to define rewards..." does not have a lot of content and could probably be cut or shortened
- Footnote on page 3: "baring the issue" -> "barring the issue"
- Section 3, discussion on theory "We provide a more complete discussion..." it would be nice to have some specifics in the main text in the form of conclusions, otherwise this takes space and distracts from the main argument
- Algorithm 1, define f_i vs. f_{w_i}
- Algorithm 1, define objective for q_w as well as for f_{w_i}
- Section 5.1, "hard exploration environments" is a subjective claim and probably deserves a citation, I don't consider these to be hard exploration environments, but maybe I should and I don't know why
- Section 5.2, "For Mountain Car, w ereport" -> "For Mountain Car, we report"

**Questions:**

- see Weaknesses above, mostly I want some more clarity around the exact objective for q_w vs. f_{w_i}
- also in Weaknesses, how does performance vs. RND and ACB change with more frames

---

> ### Author Response · Authors · 2023-11-16
>
> We thank the reviewer for a detailed review of our work and the useful suggestions. We have incorporated the fixes in the minor comments and a few other fixes based on the other comments. We address the questions you have raised below.
>
> The Atari experiments are actually run for 4 Million frames instead of 400K. The notation 1e6 is meant to demonstrate 10^6, we will make this more clear in the updated paper.
>
> RND and ACB are much more expensive than VBE and take a lot longer to run. But, we are currently completing results for these algorithms for 10 Million frames and will update the paper with these complete results.
>
> You bring up a reasonable point about the worse performance of VBE in Pitfall and Gravitar. In Pitfall our understanding is that VBE takes longer to converge to the policy of always going left in Pitfall, as the agent still tries to explore novel states. In Figure 7 in the Appendix, both variants of VBE continue to improve in Pitfall after 4 Million frames. VBE-CNN fully converges to the policy of always going left and outperforms all baselines.
>
> For Gravitar, both ACB and RND start off at a better policy than VBE. This may be due to different architectures, since ACB and RND use PPO, since that is originally how they were specified. We cannot say for sure, of course, but could put some discussion on our speculations and label them clearly as speculations (One of the reasons we chose to do a more careful study in classic control environments is precisely because we can better understand the algorithms.) It is interesting to note, though, that VBE does slowly improve, whereas ACB/RND do not. This is especially evident in Figure 7 in Appendix where both variants of VBE continue to improve beyond 4 Million frames and eventually arrive at a similar mean performance to ACB/RND in Gravitar.
>
> We did not include Montezuma’s Revenge because initial experiments indicated no baseline does well in this environment within the given frame budget (also reported in [6]). It is note-worthy that [6] also characterizes Montezuma’s Revenge as obfuscating the exploration abilities of reward-bonus based exploration methods. In the original RND work, the agent gets to use multiple parallel copies of the environment and sees much more frames (as the reviewer correctly points out), which is much much more interaction. For the sparse reward setting, we do have Sparse Mountain Car, which we have found to be surprisingly challenging for deep RL agents and actually highlights differences between methods. We will keep Antmaze in mind for future work!
>
> For Algorithm1, we can see how it was unclear how we updated q_w. Line 10 says we update using Equation 1, with the comment in line 9 saying this update is the action-values without explicitly mentioning q_w. We will fix this and update the paper soon. The q_w is updated using the standard DDQN update using the environment reward (namely, Equation 1), with no change from the DDQN update. The primary change is the behavior policy, which now uses value bonuses, and how we update the ensemble to define the value bonus.
>
> Riverswim and Deepsea [1, 2, 3, 4 ,5] are considered hard exploration environments, and are commonly used in literature to test exploration algorithms.
>
> References:
>
> [1] Kumaraswamy, R., Schlegel, M., White, A., & White, M. (2018). Context-dependent upper-confidence bounds for directed exploration. Advances in Neural Information Processing Systems, 31.
>
> [2] Ishfaq, H., Cui, Q., Nguyen, V., Ayoub, A., Yang, Z., Wang, Z., ... & Yang, L. (2021, July). Randomized exploration in reinforcement learning with general value function approximation. In International Conference on Machine Learning (pp. 4607-4616). PMLR.
>
> [3] Osband, I., Doron, Y., Hessel, M., Aslanides, J., Sezener, E., Saraiva, A., ... & Van Hasselt, H. (2019). Behaviour suite for reinforcement learning. arXiv preprint arXiv:1908.03568.
>
> [4] Ian Osband, John Aslanides, and Albin Cassirer. Randomized Prior Functions for Deep Reinforcement Learning. NeurIPS, 2018.
>
> [5] Ian Osband, Benjamin Van Roy, Daniel J Russo, Zheng Wen, et al. Deep exploration via randomized value functions. J. Mach. Learn. Res., 20(124):1–62, 2019.
>
> [6] Taiga, A. A., Fedus, W., Machado, M. C., Courville, A., & Bellemare, M. G. (2021). On bonus-based exploration methods in the arcade learning environment. arXiv preprint arXiv:2109.11052.

---

> > ### Comment · Reviewer_pttX · 2023-11-20
> > **Thanks for your response**
> >
> > Thanks for responding in detail to the suggestions.
> >
> > One very significant point that I missed on my first read of the paper (caught by reviewer E6HL) is that the likely strongest baseline RND used PPO, while your method was DQN-based, and that there were value ensemble size differences. This is a critical point when it comes to sample-efficiency and there is no reason RND needs to be tied to PPO.
> >
> > PPO can often be orders of magnitude worse in terms of sample efficiency, simply because we discard transitions, which helps the sample-efficiency of VBE by comparison. I still think that the method is capable of beating RND, but the comparison is currently very unfair.
> >
> > > We did not include Montezuma's revenge...
> >
> > I think this is fine justification. I should note: I would think parallel copies of the environment vs. single-thread but off-policy is not critical. What is critical is the rate of data collection to update ratio, and the longevity of that data.
> >
> > > For Algorithm 1, we can see...
> >
> > Thanks for the clarification. It seems this still isn't fixed in the paper though.
> >
> > > Riverswim and Deepsea
> >
> > Thanks for the additional justification.
> >
> > I am very much excited by this work's potential, but due to the critical issue in evaluation that I discussed at the beginning, I unfortunately believe it is correct for me to downgrade my score from 8 to 6. I hope the authors keep pursuing this in the future, as fixing this issue is not a huge change to make for baselines which should quickly yield results.

---

> > > ### Author Response · Authors · 2023-11-21
> > >
> > > Thank you for your response! We agree that RND does not need to be tied to PPO, and that comparing a DDQN-based algorithm with a PPO-based algorithm may not be fair. However, we did test versions of both RND and ACB when used with DDQN as the base algorithm instead of PPO. We used the same ensemble-sizes and bonus-scales for sweeps and reported best performing results. We tested these baselines on the four classic control environments (Section-5.5 & Figure-3), and in the pure exploration setup (Section-5.3 & Figure-1), where it is easier to interpret results. We can clearly see that both value-based methods (ACB & RND) struggle in the pure exploration experiment due to their inherent limitations of not providing first-visit optimism. VBE also outperforms the DDQN variants of ACB and RND on the four classic environments. It is only in the Atari experiments where we use PPO-based versions of ACB and RND. This choice was made to be fair to the baselines in the sense that we compare our method with baselines as they were originally proposed and evaluated on Atari. We will try to include an experiment with the DDQN variant of RND on Atari, but we hope that your justification makes sense. Thank you!

---

> > > > ### Comment · Reviewer_pttX · 2023-11-22
> > > >
> > > > I see, it seems I was quite confused by the distinctions in settings in the text. I see that this makes the results in Figures 1 and 3 strong, while 2 and 4 are more questionable.
> > > >
> > > > I think part of my confusion may have stemmed from the fact that RND and ACB are introduced as using PPO in Section 5.2, while VB RND and VB ACB are introduced at the end of the subsection, but the results for VB RND and VB ACB are presented earlier than for RND and ACB.
> > > >
> > > > It is understandable to use baselines in their original setting, but I feel it is somewhat common knowledge that pure policy gradient methods are much more sample inefficient than DQN-based methods. It's quite important that future revisions include VB RND/ACB for Figure 4, as those are the toughest domains.

---

> > > > > ### Author Response · Authors · 2023-11-23
> > > > >
> > > > > Thank you for your comment. We have updated the paper and have incorporated your feedback! The Atari result in the main text (Section-5.6, Figure-4) is now on 12 million steps (previously 4 million). Also we make the distinction between the PPO-based ACB/RND and DDQN-based ACB/RND (VB ACB/RND) early on and more clearly in the experiment section-5. We also added a new result in the Appendix E where we compare VBE with VB ACB/RND on the six Atari environments. The reviewer was right in noting that DDQN-based version would be more sample-efficient compared to their PPO-based counterparts. However, VBE still performs better in general, outperforming in 3/6 environments. In Qbert VBE performs better than ACB, and in Pitfall VBE performs better than RND. We also note that beyond 5 million frames on Pitfall, VBE surpasses VB ACB too (will update the paper). It is only in Gravitar where both VB ACB/RND are better than VBE, as VBE is slow but continues to improve beyond 5 million frames. We will update the paper with results on more frames in the final version, and perhaps even add it to the main text as soon as we have the results for 12 million frames for consistency.
> > > > >
> > > > > We thank the reviewer for giving thorough feedback on how to improve the paper. We hope that we have successfully incorporated the feedback in our revision and have satisfied important concerns raised by the reviewer!

---

### Official Review · Reviewer_VBsA · 2023-11-01

**Soundness:** 1 poor
**Presentation:** 1 poor
**Contribution:** 2 fair
**Rating:** 3
**Confidence:** 4

**Summary:**

This paper introduces a new approach for exploration in RL. The proposed method generates random action-value functions (RQFs) to define consistent rewards. Specifically, it asks the agent to act greedily according to $\operatorname{argmax}_a q(s, a)+b(s, a)$, where $b(s, a)$ corresponds to the value bonuses in VBE.

**Strengths:**

- Originality: The paper attempts to address exploration in reinforcement learning by introducing the Value Bonuses with Ensemble Errors (VBE). The use of random action-value functions (RQFs) to determine consistent rewards represents a departure from conventional ensemble-based methods in deep reinforcement learning.

- Quality: While there are areas in need of further clarity, the paper provides some mathematical formulations, particularly around the stochastic ensemble reward, suggesting an effort to ground the approach in theoretical foundations.

- Significance: The attempt to distinguish their method from Bootstrapped DQN shows an effort to position the paper within the broader context of ensemble-based methods in reinforcement learning. The idea of leveraging ensemble errors for deep exploration is a direction that might be worth further exploration in the future, even if this paper's execution might not fully capture the potential of the idea.

In sum, while the paper has its challenges, there is merit in the core idea it attempts to present and its potential implications for ensemble-based methods in reinforcement learning.

**Weaknesses:**

This paper describes a simple idea in a somewhat convoluted manner. Here are specific areas of concern:

- Clarity and Presentation: The paper tends to obfuscate what could be explained more simply. While there is value in rigorous mathematical explanations, these should be accompanied by intuitive explanations and clearer definitions for broader accessibility. For example, the distinction between equation 2 and the actual bonus used in algorithm 1 are not clearly demarcated, leading to potential confusion.

- Novelty Concerns: Upon close examination, the proposed method seems to be essentially a variant of the classical UCB exploration strategy, replacing the count based bonus to ensemble error based bonus; as well as a variant to RND, with difference in ensembles. Moreover, there are clear parallels with the bootstrapped DQN approach, which already encourages first-visit optimism. While the paper does acknowledge the connection to bootstrapped DQN, the explanation is not well-presented, and readers may find it challenging to discern the true novelty of the proposed method.

- Significance of Contribution: Given the above, one might argue that the paper's primary contribution is an amalgamation of previously explored ideas. The ensemble error as an exploration bonus, though interesting, might not be substantial enough to warrant a separate methodology, especially given the similarities to existing methods.

- Experimental Validation: Although not explicitly discussed earlier, it would be essential for such a paper to provide comprehensive experimental results to validate its claims, especially when the theoretical distinction from existing methods is subtle. Without this, it's challenging to gauge the real-world efficacy of the proposed approach.

For the paper to better achieve its goals, it would benefit from a clearer exposition, more transparent delineation of its novelty compared to existing methods. I am willing to raise my score if the author may improve the writing/presentation for better clarity.

**Questions:**

Please address the concerns in the weaknesses part.

---

> ### Author Response · Authors · 2023-11-16
>
> We thank the reviewer for their review, and appreciate suggestions to make the work clearer, since we are always striving to present the idea in the simplest way possible. We agree with the reviewer that our proposed methodology is an extension of UCB-style exploration methods to the RL setting. However, our proposed methodology is novel and different from existing attempts to extend UCB to the RL setting – which we refer to as reward bonus methods – like count-based methods, RND, ACB, ICM, etc. We will attempt to further highlight this distinction here. First, it is important to note that UCB-style exploration methods originate from the bandit literature, where the agent maintains and updates its reward estimates for each arm and UCB-based exploration methods provide reward bonuses to encourage selecting arms with higher uncertainty. In RL, an agent maintains and updates its value estimates to determine its behavior policy, thus an exploration bonus is required in the value space to encourage visiting uncertain states. The existing algorithms that attempt to extend such UCB-style exploration methods to the RL setting first approximate a reward bonus using counts, density-based counts, prediction errors [RND, ACB, ICM], etc., and then propagate these reward bonuses (or local estimates of uncertainty) either by adding this reward bonus to the environment reward or by updating a separate value function as in RND, ACB. These existing reward bonus methods attempt to extend UCB-style methods to RL by doing two steps: 1- approximating reward bonus or local uncertainty via supervised learning or self-supervised learning, 2- propagating these reward bonuses, say using TD-learning.
>
> Our proposed method VBE attempts to extend UCB-style methods to the RL setting by directly approximating bonuses in the value space (or value bonuses). It avoids the two step process of reward bonus approaches, that require estimating local uncertainties and then propagating. VBE directly updates the value functions in the ensemble to get a value bonus. Further, directly defining the bonuses in the value space provides first-visit optimism, which the existing reward bonus methods fail to provide. This is evident from our pure exploration experiment in Section-5.3 and Figure-1.
>
> To further understand how our value bonus method provides optimism by implicitly conserving uncertainty in the value estimates consider this example: Consider the following transition sequence: $(s, a, s’, a’)$, $f_{rqf}(s,a)$ is the predictor RQF, $f_{trg}(s,a)$ is the target RQF, and reward function for the predictor RQF is defined as $r_{rqf}(s,a,s',a') = f_{trg}(s,a) - \gamma f_{trg}(s',a')$ (Equation-3). Now suppose that the state-action pair $(s’,a’)$ has been less frequently visited and thus has high prediction error. This would mean that the error or the $bonus(s’,a’) = |f_{rqf}(s’,a’) - f_{trg}(s’,a’)|$ would be high, but how does this high uncertainty affect  $bonus(s,a)$? Notice that we update $f_{rqf}(s,a)$ by bootstrapping the prediction value of $f_{rqf}(s’,a’)$ (TD learning update), that is the target for $f_{rqf}(s,a)$ is $r_{rqf}(s,a,s’,a’) + \gamma f_{rqf}(s’,a’)$. This bootstrap target can only be the “true target”, $f_{trg}(s,a)$ if  $\gamma f_{rqf}(s’,a’) == \gamma f_{trg}(s’,a’)$, that is $f_{rqf}(s’,a’)$ is error-free. This means that prediction error for $(s,a)$ or the $bonus(s,a)$ will only go down to zero when the prediction error for all subsequent state-action pairs is zero. Proposition 1 reflects this property.
>
> Connection to BDQN explained: BDQN is not a UCB-style exploration method, rather BDQN aims to approximate Thompson Sampling for exploration in RL. The ensemble in BDQN aims to approximate the posterior distribution over action-value functions. Statistical bootstrapping alone however can only approximate the epistemic uncertainty of observed state-action pairs as noted in [1]. Adding a fixed additive prior can provide the first-visit optimism. Notice this explanation from the BDQN paper “The crucial element is that when a new state s’ is reached there is some ensemble member that estimates $\max_{a’} Q_{k}(s’, a’)$ is sufficiently positive to warrant visiting, even if it causes some negative reward along the way” [1]. The ensemble in BDQN is thus responsible for approximating the posterior distribution and providing first-visit optimism.
>
> VBE on the other hand is less sensitive to the size of its ensemble. This is especially prominent from Table-1 in Appendix B which shows that in the pure exploration experiment (Section-5.3, Figure-1) VBE only requires 1 RQF to cover the entire state-space for all grid-sizes, whereas BDQN requires 20 heads in the ensemble and still does not manage to cover the state-space for grid-sizes bigger than 35x35. This further highlights the efficacy of VBE over BDQN as it requires smaller ensembles even on hard exploration tasks and also highlights the difference in usage of the ensemble.

---

> > ### Author Response · Authors · 2023-11-16
> >
> > We summarize and highlight some advantages of VBE over existing reward bonus methods:
> >
> > 1- Defining the value bonuses directly allows us to incorporate first-visit optimism.
> >
> > 2- Single step optimization problem compared to solving two optimization problems in conventional reward bonus methods.
> >
> > 3- The reward bonus in existing methods is non-stationary, whereas the reward function for RQFs defined in Equation-3 is stationary.
> >
> > 4- Easier to combine with value-based approaches, as it is completely independent. Whereas in reward bonus methods there is a choice to be made on how to combine the reward bonus with the main value function. For example ACB and RND use a separate value function because the intrinsic value function needs to be non-episodic.
> >
> > We ask the reviewer to go through this response and identify if this explanation helps with the clarity. We are more than happy to incorporate clarifications in the paper, if you can provide more specific issues given this additional explanation. Please note  that we will fix the highlighted discrepancy with the definition of Equation 2 and the use in Algorithm 1, thank you for the pointer.
> >
> > References:
> >
> > [1] Ian Osband, John Aslanides, and Albin Cassirer. Randomized Prior Functions for Deep Reinforcement Learning. NeurIPS, 2018.

---

### Official Review · Reviewer_MrVZ · 2023-11-09

**Soundness:** 3 good
**Presentation:** 3 good
**Contribution:** 2 fair
**Rating:** 5
**Confidence:** 3

**Summary:**

This paper aims to conduct exploration over states, which is typically not done with value based exploration methods from previous works. The authors provide justifications on the convergence of their proposed VBE method and conduct experiments to show that VBE outperforms SOTA algorithms on standard testing environments.

**Strengths:**

The paper is well written with clear motivation and discussion on the relationship between VBE and BDQN. The proposed algorithm is novel to me and is interesting. Experimentation also shows that proposed VBE performs better than SOTA algorithms.

**Weaknesses:**

Regarding the claim that the proposed bonus “ensures that bonus goes to zero” when environment is sufficiently explored. In other UCB-stype work and the BDQN setup, theoretically, bonus will also goes to zero if actions are sufficiently explored.

Overall I find this work interesting but contribution is relatively marginal, given existing algorithms including BDQN, RND, ICM [1], numerous self-supervised exploration method of this style (e.g., [1][2], to name a few), and numerous theoretical analysis on UCB-styled exploration (e.g, [1] and plenty of follow-up works). In fact, [3] explicitly show that the proposed bonus function scales with an upper confidence bound in the linear setup (Lemma 4.3 of [3]).


[1] D. Pathak, et al.,  Curiosity-driven Exploration by Self-supervised Prediction. (2017)

[2] Y. Burda et al., Large-scale study of curiosity-driven learning. (2019)

[3]  Cai et al., Provably Efficient Exploration in Policy Optimization. (2020)

**Questions:**

See weaknesses.

---

> ### Author Response · Authors · 2023-11-16
>
> We thank the reviewer for their time and review and hope to address the points raised. The primary concern seems to be that this contribution is marginal, but we believe there was a misunderstanding about what our approach does and how it differs from previous work. There is no doubt that there are many approaches trying to extend UCB-style exploration to RL, namely those trying to estimate uncertainty and taking actions according to approximate upper confidence bounds. The many reward bonus approaches can absolutely be seen to do this, as we point out in the introduction. We say “One simple approach is to separate out the reward bonuses and learn them with a second value function, as was proposed for RND (Burda et al., 2019) and later adopted by ACB (Ash et al., 2022). This approach, however, still suffers from the fact that reward bonuses are only retroactive, and the resulting b is unlikely to be high for unvisited states and actions.” We demonstrate this in the pure exploration experiment Section-5.3, Figure-1, where the reward bonus methods fail to cover the state space as these methods do not encourage visiting novel states.
>
> The reviewer mentions several reward bonus methods like RND, ICM, which aim to extend UCB-style exploration methods to the RL setting. We will further try to clarify the difference between our approach and existing reward bonus methods here. RND uses a fixed randomly initialized network to define the target for a state, and a separate prediction network is trained to predict that target using a standard supervised learning MSE loss. The reward bonus for a state is defined as the prediction error for that state. This reward bonus can not be directly used to alter the agent’s behavior as the prediction error only approximates local epistemic uncertainty. To do directed deep exploration RND uses this bonus as a reward to update a separate value function to propagate local uncertainties to earlier states. Now, since we have the uncertainty in value space, the agent’s behavior can be altered to do deep exploration. Most reward bonus methods follow this exact principle, with the difference being in how local uncertainties are estimated. For example, ACB uses stochastic targets and the bonus is defined as the maximum deviation from the mean over an ensemble. ICM uses prediction errors in the forward dynamics model. All such reward bonus methods use a two-step process, 1- measure local uncertainties (prediction errors, counts, variance, etc.), 2- propagate local uncertainties to get the value bonus (uncertainty estimate in value space).
>
> Our proposed methodology is different from these reward bonus methods in that it aims to approximate the value bonus directly in one step, by updating the RQFs using the reward function defined for each RQF (Equation-3) and measuring the error in predictions w.r.t. true target RQFs. In doing so we also solve the problem of first-visit optimism which is a known problem with reward bonus methods.
>
> The other claim is around theoretical analysis. To the best of our knowledge, there is very little theory in deep RL for upper confidence bound approaches (and value bonus approaches). The reviewer states that [1] has theory on this topic, but that work is focused on empirically understanding a proposed intrinsic reward (reward bonus). As mentioned above, such reward bonus methods fail to incorporate first visit optimism, as the intrinsic reward only affects the value function after visiting a state-action pair. The paper [3] proposes OPPO, an optimistic version of PPO. It is important to note that this methodology and the theory is only applicable in the fixed horizon, linear approximation setting.  Extending this theory to the deep RL setting, for episodic problems that have variable length episodes, remains an open question. We did cite several theory works in the introduction; we will include this paper in that list.
>
> When we say our bonuses go to zero by design, we are not saying that other UCB-style algorithms have bonuses that do not go to zero. We are simply pointing out that this desirable property is also a property of our algorithm.

---

### Meta-Review · Area_Chair_xxVD · 2023-12-05

**Metareview:**

Summary: The work introduces a novel methodology, Value Bonuses with Ensemble Errors (VBE), for extending UCB-style exploration to the RL setting. Authors use an ensemble of Q-functions to provide optimistic value estimates for unvisited state-action pairs. Unlike previous methods that rely on retroactive exploration bonuses, VBE offers a mechanism for exploration that rewards states never visited through uncertainty in the ensemble of Q-functions.

Weaknesses: Reviewers have raised concerns about the contribution's novelty, especially compared to existing methods extending UCB-style exploration to RL, clarity in presentation, and fair evaluation of baselines. Concerns are also raised about the experimental design, choice of environments, and comparisons between different reinforcement learning algorithms (Double Deep Q-Network (DDQN) vs. Proximal Policy Optimization (PPO)). While authors did add experiments with a fairer comparison to baselines, results are less significant.

Strengths: The paper proposes novel exploration bonuses and the method's sample efficiency shows improvement over baselines.

Suggestions: Add experiments in harder exploration scenarios. Just because papers with certain environments were accepted a few years ago is not a good justification for not attempting harder environments. Benchmarks should improve as our methods do.

Note for the future: when uploading revisions of your paper, please color code the changes.

**Justification For Why Not Higher Score:**

Clarity in presentation could be improved. Concerns are also raised about the experimental design, choice of environments, and comparisons between different reinforcement learning algorithms (Double Deep Q-Network (DDQN) vs. Proximal Policy Optimization (PPO)). While authors did add experiments with a fairer comparison to baselines, results are less significant.

**Justification For Why Not Lower Score:**

N/A

---

### Decision · Program_Chairs · 2024-01-16

Reject